# Analysis of Nonlinear Convection–Radiation in Chemically Reactive Oldroyd-B Nanoliquid Configured by a Stretching Surface with Robin Conditions: Applications in Nano-Coating Manufacturing

**DOI:** 10.3390/mi13122196

**Published:** 2022-12-11

**Authors:** Muhammad Nasir, Muhammad Waqas, O. Anwar Bég, Hawzhen Fateh M. Ameen, Nurnadiah Zamri, Kamel Guedri, Sayed M Eldin

**Affiliations:** 1Faculty of Informatics and Computing, Besut Campus, University Sultan Zainal Abidin, Besut 22200, Malaysia; 2NUTECH School of Applied Science and Humanities, National University of Technology, Islamabad 44000, Pakistan; 3Department of Mechanical Engineering, Lebanese American University, Beirut 1102, Lebanon; 4Mechanical Engineering, School of Science, Engineering and Environment (SEE), Salford University, Manchester M5 4WT, UK; 5Department of Petroleum Engineering, College of Engineering, Knowledge University, Erbil 44001, Iraq; 6Mechanical Engineering Department, College of Engineering and Islamic Architecture, Umm Al-Qura University, P.O. Box 5555, Makkah 21955, Saudi Arabia; 7Center of Research, Faculty of Engineering, Future University in Egypt, New Cairo 11835, Egypt

**Keywords:** nonlinear mixed convection, stagnant-point flow, stretching permeable surface, Oldroyd-B nanoliquid, chemical reaction, nonlinear thermal radiation, homotopy analysis scheme, nano-coating fabrication

## Abstract

Motivated by emerging high-temperature manufacturing processes deploying nano-polymeric coatings, the present study investigates nonlinear thermally radiative Oldroyd-B viscoelastic nanoliquid stagnant-point flow from a heated vertical stretching permeable surface. Robin (mixed derivative) conditions were utilized in order to better represent coating fabrication conditions. The nanoliquid analysis was based on Buongiorno’s two-component model, which features Brownian movement and thermophoretic attributes. Nonlinear buoyancy force and thermal radiation formulations are included. Chemical reactions (constructive and destructive) were also considered since coating synthesis often features reactive transport phenomena. An ordinary differential equation model was derived from the primitive partial differential boundary value problem using a similarity approach. The analytical solutions were achieved by employing a homotopy analysis scheme. The influence of the emerging dimensionless quantities on the transport characteristics was comprehensively explained using appropriate data. The obtained analytical outcomes were compared with the literature and good correlation was achieved. The computations show that the velocity profile was diminished with an increasing relaxation parameter, whereas it was enhanced when the retardation parameter was increased. A larger thermophoresis parameter induces an increase in temperature and concentration. The heat and mass transfer rates at the wall were increased with incremental increases in the temperature ratio and first order chemical reaction parameters, whereas contrary effects were observed for larger thermophoresis, fluid relaxation and Brownian motion parameters. The simulations can be applied to the stagnated nano-polymeric coating of micromachines, robotic components and sensors.

## 1. Introduction

Boundary-layer flows that are configured by moving surfaces have remarkable industrial applications, including fiber technology, the aerodynamic extrusion of plastic sheets, melting spinning, the enhanced recovery of petroleum resources, artificial fibers, hot rolling, glass-fiber and paper manufacturing and the production of polythene items. The theory of boundary layer flow, which is caused by a continuously flat surface moving with constant velocity, was first described by Sakiadis [1]. Crane [2] modeled boundary-layer stretching flow and computed closed-form solutions. Various researchers have extended the analyses of Sakiadis [1] and Crane [2] under diverse physical aspects. For instance, Govindaraj et al. [3] presented 2D water type boundary-layer flow confined by a moving exponential vertical surface and analyzed the impact of Prandtl number and variable viscosity. Ibrahim and Gadisa [4] explored the aspects of suction/injection in micropolar nanoliquid boundary-layer flow based on improved theories of heat–mass transfer. Rajput et al. [5] inspected the effectiveness of a porous medium on magnetohydrodynamic boundary-layer flow across an exponentially stretched surface and solved it numerically by applying the shooting technique. The phenomenon of boundary-layer flow over a vertical Riga plate with magnetohydrodynamics (MHDs) and viscous dissipation has been carried out by Eldabe et al. [6]. Dawar et al. [7] explored mixed convective boundary-layer flow featuring chemically reactive micropolar liquid over a stretchable surface with Joule heating. Nadeem et al. [8] numerically described 2D boundary-layer flow considering the Buongiorno nanoscale model past a nonlinear extending surface and captured the impacts of porous mediums. Further analyses in this direction are mentioned in references [9,10,11,12,13,14,15,16,17,18,19,20].

At present, researchers are scrutinizing viscoelastic liquids due to their broad engineering and industrial applications. These liquids are used in prescribed medications [21], colloidal matter, physiological manufacturing, the manufacturing of volatiles, food manufacturing [22,23], oil reservoir engineering, fabric technologies, melts of polymeric, thermal circuit processing [24,25] and chemical/nuclear industries [26]. Stress and strain are not enough to distinguish between the different characteristics of these liquids. For diverse rheological characteristics, numerous constitutive relationships have been developed. In the current study, the Oldroyd-B liquid model was considered, which expands on the effects of memory and elasticity which are common features in many biological and polymeric liquids deployed in modern coating systems. In addition, this model also captures the effects of relaxation and retardation time. This model reduces to the Maxwell fluid in the absence of retardation time. Furthermore, when both relaxation and retardation times are omitted, the Oldroyd-B model retracts to the classical (viscous fluid) Newtonian model. This model has been implemented by various researchers under diverse flow assumptions. For example, Hafeez et al. [27] employed the bvp4c scheme design using MATLAB software in order to compute numerical solutions for Cattaneo–Christov-based Oldroyd-B nanoliquid towards a spinning disc. Rana et al. [28] presented a mathematical analysis of Oldroyd-B nanofluids configured by permeable surfaces considering gyrotactic microorganisms. The bioconvection impact in Oldroyd-B liquid slip flow configured by a heated surface was evaluated by Khan et al. [29]. They concluded that the solutal profile decreases with an increasing Schmidt number and retardation time constant. Ali et al. [30] considered the Soret–Dufour and MHD effects in Cattaneo–Christov heat–mass flux models based on Oldroyd-B nanoliquid rotating flow employing the finite element scheme. The significance of the Robin conditions in chemically reactive radiated Oldroyd-B nanoliquid, which considers the thermophoretic and Brownian diffusion effects, was discussed by Irfan et al. [31]. They noted that solutal and thermal Biot numbers have the same influence on concentration and temperature fields, respectively. Yasir et al. [32] computed homotopic convergent solutions for Oldroyd-B liquid stretched flow that was subjected to energy transport. Recent studies related to Oldroyd-B fluid are reported in references [33,34,35,36,37].

This analysis aims to extend the research presented by Nasir et al. [38] for nonlinear radiative Robin conditions for Oldroyd-B nanoliquids confined by a stretchable porous vertical surface. Nonlinear mixed convective flow in the stagnation region was formulated. The mass transfer effects were explored considering the chemical reactions. Undoubtedly, various computational (analytical and numerical) schemes [39,40,41,42,43,44,45,46,47] are available in the existing literature. Here, the nonlinear problems were computed using a homotopic approach [48,49,50] for convergent series solutions. Moreover, graphical depictions of some of the key findings with a detailed discussion have also been incorporated. The applications of the present non-Newtonian nanofluid stagnation model includes coating manufacturing processes for biomimetic sensors [51], optical fiber nanocoatings [52,53] and micro-robot surface protection [54,55]. The key objective of the present study was to simultaneously consider multiple effects which feature in real manufacturing stagnation flows for nanopolymer coatings, including complex thermal convective wall boundary conditions, rheology, high temperatures (radiative flux), Brownian motion and thermophoresis. These have not been addressed previously in the Oldroyd-B viscoelastic model. 

## 2. Mathematical Model

We considered nonlinear mixed convective flow in the stagnation region of the Oldroyd-B nanoliquid confined by a heated permeable vertical surface as a model for the deposition of a nanopolymer in the manufacturing of coatings. The rheological nature of many polymeric nanocoatings requires a robust non-Newtonian model that can simulate real effects including relaxation and retardation, for which the Oldroyd-B model is an excellent candidate. The nanoliquid model presented by Buongiorno [8], featuring Brownian diffusion and thermophoresis effects, was utilized since it provides a two-component (thermosolutal) framework for analysis. The nonlinear radiative aspects, along with the Robin conditions, was considered to represent high temperature manufacturing conditions and complex wall conditions in coating deposition processes. The mass transfer effects were explored considering the chemical reactions, which is also common in the synthesis of nanomaterial polymeric coatings. The physical model is visualized in Figure 1. The governing equations under the considered effects are [34]:
(1)$$\frac{\partial u}{\partial x}+\frac{\partial v}{\partial y}=0,$$
(2)$$\left\{ \begin{array}{l} u\frac{\partial u}{\partial x}+v\frac{\partial u}{\partial y}+\lambda_{1}\left(u^{2}\frac{\partial^{2}u}{\partial x^{2}}+v^{2}\frac{\partial^{2}v}{\partial y^{2}}+2uv\frac{\partial^{2}u}{\partial x\partial y}\right)=\nu \left(\frac{\partial^{2}u}{\partial y^{2}}+\lambda_{2}\left(\frac{\partial^{3}u}{\partial x\partial y^{2}}u-\frac{\partial u}{\partial x}\frac{\partial^{2}u}{\partial y^{2}}-\frac{\partial u}{\partial y}\frac{\partial^{2}v}{\partial y^{2}}+v\frac{\partial^{3}u}{\partial y^{3}}\right)\right)\\ +u_{e}\frac{du_{e}}{dx}+g\left\{\Lambda_{1}\left(T-T_{\infty }\right)+\Lambda_{2}{\left(T-T_{\infty }\right)}^{2}\right\}+g\left\{\Lambda_{3}\left(C-C_{\infty }\right)+\Lambda_{4}{\left(C-C_{\infty }\right)}^{2}\right\}, \end{array}\right.\;$$
(3)$$u\frac{\partial T}{\partial x}+v\frac{\partial T}{\partial y}=\alpha \frac{\partial^{2}T}{\partial y^{2}}+\tau \left[D_{B}\frac{\partial C}{\partial y}\frac{\partial T}{\partial y}+\frac{D_{T}}{T_{\infty }}{\left(\frac{\partial T}{\partial y}\right)}^{2}\right]+\frac{16\sigma^{*}}{3k^{*}{\left(\rho c\right)}_{f}}\frac{\partial }{\partial y}\left(T^{3}\frac{\partial T}{\partial y}\right),\;$$
(4)$$u\frac{\partial C}{\partial x}+v\frac{\partial C}{\partial y}=D_{B}\frac{\partial^{2}C}{\partial y^{2}}+\frac{D_{T}}{T_{\infty }}\frac{\partial^{2}T}{\partial y^{2}}-K_{1}\left(C-C_{\infty }\right)\;$$
(5)$$\begin{array}{l} u=u_{w}(x)=cx,v=v_{w},\;\;-k\frac{\partial T}{\partial y}=h_{f}\left(T_{f}-T\right),\;-D_{B}\frac{\partial C}{\partial y}=h_{c}\left(C_{f}-C\right) \end{array}\mathrm{at}\;y=0,\;$$
(6)$$u\to u_{e}(x)=ex,\;T\to T_{\infty },\;C\to C_{\infty }\;\mathrm{when}\;y\to \infty .\;$$

Here $\nu \left(=\frac{\mu }{\rho_{f}}\right)$ denotes the kinematic viscosity of the sheet, $\rho_{f}$ denotes liquid density, μ denotes dynamic viscosity, $\lambda_{1}$ denotes relaxation time, k denotes thermal conductivity, $u_{e}\left(x\right)$ denotes free stream velocity, g denotes gravitational acceleration, $T_{\infty }$ denotes the ambient temperature of the liquid, $D_{T}$ denotes the thermophoresis diffusion coefficient, $\left(\Lambda_{1},\Lambda_{3}\right)$ denote the linear (thermal, concentration) expansion coefficients, the heat capacity ratio is represented by $\tau =\frac{{\left(\rho c\right)}_{P}}{{\left(\rho c\right)}_{f}}$ with liquid heat denoted by ${\left(\rho c\right)}_{f}$, the nanoparticle effective heat capacity is denoted by ${\left(\rho c\right)}_{p}$, $\sigma^{*}$ denotes the Stefan–Boltzmann constant, $D_{B}$ denotes the Brownian diffusion coefficient, $\lambda_{2}$ denotes retardation time, $K_{1}$ denotes the reaction rate, $k^{*}$ denotes the mean absorption coefficient, $\alpha =\frac{k}{{\left(\rho c\right)}_{f}}$ denotes thermal diffusivity, T denotes the liquid temperature, $\left(u_{w}(x),v_{w}\right)$ denotes stretching and suction/injection velocity, c denotes the dimensional constants, $h_{f}$ denotes the heat transfer coefficient, $\left(\Lambda_{2},\Lambda_{4}\right)$ denote the nonlinear (thermal, concentration) expansion coefficients, C denotes the liquid concentration, $h_{c}$ denotes the mass transfer coefficient, $C_{\infty }$ denotes the ambient concentration of the liquid and u, v are the elements of fluid velocity in the x, y direction, respectively. 

After implementing suitable similarity transformations [29]: (7)$$\left\{ \begin{array}{l} \eta =\;y\sqrt{\frac{c}{\nu }},\;\;u=cxf^{\prime }\left(\eta \right),\;\;v=-\sqrt{c\nu }f\left(\eta \right),\\ \theta (\eta )=\frac{T-T_{\infty }}{T_{f}-T_{\infty }},\phi (\eta )=\frac{C-C_{\infty }}{C_{f}-C_{\infty }}. \end{array}\right.$$
the continuity equation (i.e., Equation (1)) is satisfied and the other expressions in dimensionless forms are: (8)$$\begin{array}{l} f^{\prime \prime \prime }+f\;f^{\prime \prime }+\beta_{1}\left(2f\;f^{\prime }\;f^{\prime \prime }\;-f^{2}f^{\prime \prime \prime }\right)+\beta_{2}\left(f^{\prime \prime 2}-{\mathrm{ff}}^{\left(iv\right)}\right)-{\left(f^{\prime }\right)}^{2}\\ \;\;\;\;\;\;\;\;\;\;\;\;\;\;\;\;\;\;\;\;\;\;\;\;\;\;+\delta \left[\left(1+\beta_{t}\theta \right)\theta +N\left(1+\beta_{c}\phi \right)\phi \right]+A^{2}=0, \end{array}$$
(9)$$\begin{array}{l} \left(1+R\right)\theta^{\prime \prime }+\frac{4}{3}R\left[\begin{array}{l} {\left(\theta_{R}-1\right)}^{3}\left(3\theta^{2}{\left(\theta^{\prime }\right)}^{2}+\theta^{3}\theta^{\prime \prime }\right)\\ +3{\left(\theta_{R}-1\right)}^{2}\left(2\theta {\left(\theta^{\prime }\right)}^{2}+\theta^{2}\theta^{\prime \prime }\right)\\ +3\left(\theta_{R}-1\right)\left({\left(\theta^{\prime }\right)}^{2}+\theta \theta^{\prime \prime }\right) \end{array}\right]\\ \;\;\;\;\;\;\;\;\;\;\;\;\;\;\;\;+\mathrm{Pr}f\theta^{\prime }+\mathrm{Pr}\left(N_{t}{\theta^{\prime }}^{2}+N_{b}\phi^{\prime }\theta^{\prime }\right)=0, \end{array}$$
(10)$$\phi^{\prime \prime }+Sc\left(f\phi^{\prime }-\gamma \phi \right)+\frac{N_{t}}{N_{b}}\theta^{\prime \prime }=0,\;$$
(11)$$\left\{ \begin{array}{l} \mathrm{at}\;\;\eta =0,\;\;f=S,\;\;f^{\prime }=1,\;\;\phi ’=-\gamma_{2}\left(1-\phi \left(\eta \right)\right),\;\;\theta ’=-\gamma_{1}\left(1-\theta \left(\eta \right)\right)\;,\;\;\\ \;\;\;\;\;\;\;\;\;\;\;\;\;\;\;\;\;\;\;\;\;\;\;\;\;\;\;\;\;\;\;\;\;\;\;\;\mathrm{as}\;\;\eta \to \infty ,\;\;f^{\prime }\to A,\;\;\;\phi \to 0\;,\;\;\;\theta \to 0\;\;. \end{array}\right.\;$$

Here $\left({}^{\prime }\right)$ signifies differentiation concerning η, λ denotes the mixed convection parameter, Sc denotes the Schmidt number, $Gr_{x}$ denotes the thermal buoyancy number, $\beta_{1}$ denotes the dimensionless relaxation time parameter, γ<0 denotes the generative reaction variable, $Gr_{x}^{*}$ denotes the concentration buoyancy number, $\beta_{2}$ denotes the dimensionless retardation time parameter, $\beta_{t}$ denotes the nonlinear thermal convection parameter, $N_{t}$ denotes the thermophoresis parameter, $\theta_{R}$ denotes the temperature ratio parameter, $\beta_{c}$ denotes the nonlinear concentration convection parameter, Pr denotes the Prandtl number, $N_{b}$ denotes the Brownian motion parameter, R denotes the radiation variable, A denotes the ratio of rates, $\gamma_{1}$ denotes the thermal Biot number, N denotes the buoyancy ratio parameter, $\gamma_{2}$ denotes the concentration Biot number, γ>0 denotes the destructive reaction variable, (S>0) denotes suction and (S<0) denotes injection.

These parameters are defined as follows:(12)β1=cλ1, β2=cλ2,  δ= GrxRex2,  Grx= gΛ1(Tf−T∞)x3ν2,  Grx∗= gΛ3(Cf−C∞)x3ν2,βt= Λ2(Tf−T∞)Λ1,  βc= Λ4(Cf−C∞)Λ3,  N= Grx∗Grx,  Rex= xuwν,  S=vwcνNt= τDT(Tf−T∞)T∞ν,  A=  ec,   Pr=να,  Nb= τDB(Cf−C∞)ν,γ=K1c,θR=TfT∞,R= 4σ∗T∞3kk∗,  Sc=νDB,  γ1= hfkνc,  γ2= hcDBνc.

The expressions for the local Nusselt and local Sherwood numbers are:(13)$$Nu_{x}=\;{\left[-\frac{x}{\left(T_{f}-T_{\infty }\right)}\left(\frac{\partial T}{\partial y}\right)-\frac{16\sigma^{*}x}{3kk^{*}\left(T_{f}-T_{\infty }\right)}T^{3}\frac{\partial T}{\partial y}\right]}_{y=0},$$
(14)$$Sh_{x}=\;\frac{xj_{w}}{D_{B}\left(C_{f}-C_{\infty }\right)},\;\;j_{w}=\;-D_{B}{\left(\frac{\partial C}{\partial y}\right)}_{y=0}.\;$$

In non-dimensional forms, one has:(15)$$Nu_{x}Re_{x}^{-0.5}=-\left(1+\frac{4}{3}R{\left(1+\left(\theta_{R}-1\right)\theta \left(0\right)\right)}^{3}\right)\theta^{\prime }(0)\;\;\;\;\;\;\;\;\;\;\;\;\;\;\;\;\;\;\;\;\;\;\;\;\;\;\;\;\;\;$$
(16)$$Sh_{x}Re_{x}^{-0.5}=-\phi^{\prime }(0)\;$$

## 3. Analytical Solution Procedure 

Here, convergent series solutions of the nonlinear ordinary differential Equations (8) to (10), along with the boundary conditions (11), were obtained using the homotopy analysis scheme [48]. For the problem under consideration, the required initial estimations $\left(f_{o}(\eta )\;,\;\;\theta_{o}(\eta ),\;\;\phi_{o}(\eta )\right)$ and essential linear operators $\left(L_{f},\;\;L_{\theta },\;\;L_{\phi }\right)$ are specified as:(17)$$\begin{array}{l} f_{0}\left(\eta \right)=S+A*\eta +\left(1-A\right)\left(1-e^{-\eta }\right),\;\\ \;\theta_{0}\left(\eta \right)=\left(\frac{\gamma_{1}}{1+\gamma_{1}}\right)*e^{-\eta },\;\\ \;\phi_{0}\left(\eta \right)=\left(\frac{\gamma_{2}}{1+\gamma_{2}}\right)*e^{-\eta }, \end{array}$$
(18)$$\left\{ \begin{array}{l} L_{f}=f^{\prime \prime \prime }-f^{\prime },\;\\ L_{\theta }=\theta^{\prime \prime }-\theta ,\;\\ L_{\phi }=\phi^{\prime \prime }-\phi , \end{array}\right.\;$$
with
(19)$$\left\{ \begin{array}{l} L_{f}\left(A_{1}+A_{2}e^{\eta }+A_{3}e^{-\eta }\right)=0,\;\;\\ L_{\theta }\left(A_{4}e^{\eta }+A_{5}e^{-\eta }\right)=0,\;\;\\ L_{\phi }\left(A_{6}e^{\eta }+A_{7}e^{-\eta }\right)=0, \end{array}\right.\;$$
where $C_{i}\left(i=1-7\right)$ indicates the arbitrary constants.

## 4. Convergence Analysis 

The homotopic algorithm encompasses the auxiliary parameters $\hbar_{f},\;\hbar_{\theta }$ and $\hbar_{\phi }$, which efficiently control the convergence region of the homotopic solutions. These parameters ($\hbar_{f},\;\hbar_{\theta }$ and $\hbar_{\phi }$) were evaluated by plotting h-curves (Figure 2). From Figure 2, we obtained $-1.2\leq \hbar_{f}\leq -0.4,\;\;-1.2\leq \hbar_{\theta }\leq -0.2$ and $-1.2\leq \hbar_{\phi }\leq -0.2$, respectively. Table 1 shows the convergence of the homotopic solutions for velocity $f^{\prime \prime }\left(0\right)$, temperature $\theta^{\prime }\left(0\right)$, and concentration $\phi^{\prime }\left(0\right)$. It was revealed that the 25th order of approximation was sufficient for velocity, whereas temperature and concentration require only the 20th order of approximations.

## 5. Comparison of HAM Results 

Table 2 and Table 3 show the evaluation of skin friction (−CfxRex−12) with earlier simpler models, including [31,32] and [49,50]. An excellent agreement has been attained with the current homotopy solutions, as demonstrated in these tables, confirming the validity and accuracy of the present analytical results.

## 6. Results and Discussions 

This section captures the significance of the involved parameters against the dimensionless concentration, temperature and velocity profiles using Figure 3, Figure 4, Figure 5, Figure 6, Figure 7, Figure 8, Figure 9, Figure 10, Figure 11, Figure 12, Figure 13, Figure 14, Figure 15, Figure 16, Figure 17, Figure 18 and Figure 19 and Table 4. The values of the parameters were considered in the following ranges: A (0.2≤A≤1.6), $\beta_{1}$ ($0.2\leq \beta_{1}\leq 0.8$), $\beta_{2}$ ($0.2\leq \beta_{2}\leq 0.8$), δ (1.0≤δ≤7.0), N (10.0≤N≤40.0), S (−0.5≤S≤0.5), Pr (0.1≤Pr≤0.7), $N_{t}$ ($1.0\leq N_{t}\leq 4.0$), R (0.1≤R≤0.7), $N_{b}$ ($1.0\leq N_{b}\leq 4.0$), $\theta_{R}$ ($1.0\leq \theta_{R}\leq 2.5$), $\gamma_{1}$ ($0.1\leq \gamma_{1}\leq 0.4$), Sc (0.1≤Sc≤0.7), $N_{b}$ ($0.1\leq N_{b}\leq 0.4$), $N_{t}$ ($0.1\leq N_{t}\leq 0.4$), γ (−0.3≤γ≤0.3) and $\gamma_{2}\;$ ($0.1\leq \gamma_{2}\;\leq 0.4$).

### 6.1. Velocity Profile

Figure 3, Figure 4, Figure 5, Figure 6, Figure 7 and Figure 8 disclose the impact of A, $\beta_{1}$ $\beta_{2}$, δ, S and N on velocity $f^{\prime }\left(\eta \right)$. Figure 3 demonstrates that with an increase in the velocity ratio factor A*,* $f^{\prime }\left(\eta \right)$ indicates an improvement with respect to η in the boundary regime, that is, the fluid motion is accelerated on the stretchy surface. When (A>1), the free stream velocity is stronger than the stretching velocity of the surface. This builds a momentum upsurge in the flow domain through the exterior free stream which exhibits high acceleration for all horizontal coordinate values, *η*. Consequently, the thickness of the momentum boundary layer of the stretching surface decreased. This has an influence on the quality control of manufactured coatings. However, when (A<1), the stretching velocity of the sheet was higher than the exterior velocity of the free stream, and the reverse impact was calculated, i.e., the velocity of the Oldroyd-B fluid $f^{\prime }\left(\eta \right)$ decreased and the thickness of the momentum boundary layer on the surface increased. If (A=1), both the external and the stretchy velocity of the Oldroyd-B fluid were identical. This scenario is a natural intermediate between cases where *A* > 1 and *A*< 1. It was evident that a sheet with greater stretching capacity prevents the development of momentum; however, a stronger external velocity for the corresponding Oldroyd-B fluid is produced. The influence of Deborah numbers on the velocity of the Oldroyd-B fluid $f^{\prime }\left(\eta \right)$ in terms of ($\beta_{1}$ relaxation time quantity) and ($\beta_{2}\;$ retardation time quantity) are portrayed in Figure 4 and Figure 5, respectively. In Figure 4, it is clear that as $\beta_{1}$ (relaxation time fluid) rises, the Oldroyd-B fluid velocity gradually declines. In fact, $\beta_{1}$ is mathematically expressed as ‘‘the ratio of the observational timescale to the timescale of the material reaction”. We can examine the three different scenarios to assess the polymeric behavior of the materials: (i) when $\left(\beta_{1}=1\right)$ for the viscoelastic substance, (ii) when $\left(\beta_{1}<<1\right)$ for entirely viscous materials and (iii) when $\left(\beta_{1}>>1\right)$ for the elastic material in nature. Higher values of $\beta_{1}$ lead to lower relaxation relative to the characteristic timescale. This means that fluid reacts in a similar way to solid materials. Figure 5 displays the impact of the retardation time $\left(\beta_{2}=0.2,\;\;0.4,\;\;0.6,\;\;0.8\right)$ parameter on the Oldroyd-B fluid velocity $f^{\prime }\left(\eta \right)$. As anticipated, $f^{\prime }\left(\eta \right)$ upsurges when $\beta_{2}\;$ is increased. Physically, retardation time is augmented with increasing $\beta_{2}\;$, owing to the upsurges observed for $f^{\prime }\left(\eta \right)$. Figure 6 reveals the features of the mixed convection variable δ on $f^{\prime }\left(\eta \right)$. Clearly $f^{\prime }\left(\eta \right)$ increases when the mixed convection variable is elevated. Since thermal buoyancy forces exceed the viscous forces with higher values of δ, this intensifies the flow. Figure 7 unveils the impact of the buoyancy ratio variable N on $f^{\prime }\left(\eta \right)$. It can be seen that a massive boost is induced in linear velocity across the domain with stronger values of $N=\;\frac{Gr_{x}^{*}}{Gr_{x}}$. Note that N>>1 signifies that the concentration buoyancy force $Gr_{x}{}^{*}$ of the nanoparticles is much stronger than the temperature buoyancy force $Gr_{x}$. Consequently, the hydrodynamic boundary layer thickness of the Oldroyd-B fluid increases. Figure 8 depicts the characteristics of wall suction/injection on $f^{\prime }\left(\eta \right)$. The impetus of linear velocity increased with the amplification of the injection parameter (S<0). On the other hand, it decreased for greater values of wall suction factor (S>0). It was evident that the injection of nanofluid augments momentum development in the boundary layer regime and, consequently, the velocity of the Oldroyd-B nanofluid increases. This effect is also known as blowing in manufacturing processes. The reverse situation was noted for suction, which causes the boundary layer to adhere more strongly to the wall and decelerates flow. This increases the thickness of the momentum boundary layer. 

### 6.2. Temperature Profile

Figure 9, Figure 10, Figure 11, Figure 12, Figure 13 and Figure 14 display the effects of R, Pr, $\gamma_{1}$, $N_{t},$ $\theta_{R}$ and $N_{b}$ on θ(η). Figure 9 depicts the θ(η) curves subjected to Prandtl number (Pr) values. This Figure shows that the nanofluid temperature decreases for larger Prandtl numbers. The Prandtl number (Pr) denotes the rate of momentum for thermal diffusion. Liquid conductivity retards for larger values of Pr. The heat transferred via the conduction of molecules is subsequently repressed, which is represented by a decay in θ(η) and a diminution in the thickness of the thermal boundary layer. Thus, the stretched coating system is chilled with an increase in Pr, whereas heating is witnessed with a lower Pr. Figure 10 illustrates the curves of θ(η) under the impact of R. When R is increased, θ(η) is obviously enhanced. Actually, greater heat (thermal energy) is produced in the working liquid during the radiation process (i.e., for higher *R* values), causing the temperature of the nanofluid in the boundary layer regime to increase. Figure 11 shows the effects of Buongiorno’s model parameter (nanoscale thermophoresis $N_{t}$) on the thermal distribution θ(η) of the Oldroyd-B fluid. A stronger movement of nanoparticles in the stagnation boundary-layer flow domain is encouraged by the thermophoretic body force. The nanoparticles are mobilized from the hotter region to the cooler one; therefore, higher thermal transmission arises in the flow field. Hence, intensifying the magnitude of the nanoscale thermophoresis parameter $N_{t}$ produces substantial thermal diffusion enhancement and boosts temperature and thermal boundary layer thickness. The influence of the Brownian motion factor $N_{b}$ on the temperature θ(η) of the Oldroyd-B nanofluid is presented in Figure 12. There is an accentuation in both the nanofluid temperature θ(η) and the thermal boundary layer thickness as $N_{b}$ increases. In fact, the random movement of the fluid particles is increased since the nanoparticle diameters are reduced with higher Brownian motion parameters. This intensifies the ballistic collisions, which produces extra heat in the regime. Micro-convection around the nanoparticles is also enhanced with greater Brownian motion effects. All of these factors contribute to marked thermal enhancement. The effects of the temperature ratio variable $\theta_{R}$ on temperature distribution θ(η) is illustrated in Figure 13. Physically, temperature appears to rise significantly as $\theta_{R}$ increases. Higher values of $\theta_{R}$ implies an elevation in wall temperature, which makes the depth of thermal penetration deeper into the boundary layer. Heat transfer into the flow from the wall is therefore encouraged. Furthermore, when the liquid temperature $T_{f}$ exceeds the ambient temperature $T_{\infty }$ (in the energy equation) this creates a larger temperature differential across the boundary layer, which intensifies thermal diffusion from the wall to the free stream and manifests in a rise in θ(η). Figure 14 shows that with larger values of the thermal Biot number $\gamma_{1}\;$, there is an improvement in the temperature field θ(η). The condition $\gamma_{1}=0$ suggests the configuration of isoflux at the wall, whereas $\gamma_{1}\to \infty$ symbolizes the configuration of an isothermal wall. In addition, this parameter is featured in the prescribed boundary condition $\theta =-\gamma_{1}\left(1-\theta \left(\eta \right)\right)$ (from Equation (11)), which determines the intensity of the Biot number. Therefore, stronger Biot number $\gamma_{1}$ values correspond to an amplification in thermal convection over the stretching sheet, which enhances the nanofluid temperature. The inclusion of this complex convective wall boundary condition provides a more realistic estimation of the manufacturing conditions compared to the conventional boundary conditions. 

### 6.3. Concentration Profile

Figure 15, Figure 16, Figure 17, Figure 18 and Figure 19 show the impacts of Sc, γ, $N_{t},$ $\gamma_{2},$ and $N_{b}$ on ϕ(η). Figure 15 reveals that with higher values of Sc, the nanoparticle concentration ϕ(η) and the thickness of the solutal boundary layer is reduced. Actually, the Schmidt number is the ratio of momentum to the nanoparticle molecular diffusivity, which implies that when Sc  rises, mass diffusivity reduces and there is a depletion in concentration ϕ(η). The judicious selection of nanoparticles to embed in the coating regime is therefore critical in achieving bespoke mass transfer characteristics throughout fabrication processes. Figure 16 and Figure 17 portray changes in the concentration ϕ(η) of nanoparticles with different values of thermophoresis $N_{t}$ and Brownian $N_{b}$ diffusion parameters. When $N_{b}$ increases, an increasing trend is seen via higher chaotic movements which are associated with a higher $N_{b}$, the particle collision is boosted and mass diffusion is assisted. As a result, the Oldroyd-B nanofluid concentration upsurges. The reverse trend is noted for the influence of the thermophoresis parameter $N_{t}$ on the surface concentration ϕ(η). Physically, the movement of nanoparticles from the wall to the interior boundary layer region is impeded by an elevation in the thermophoretic force. Therefore, this factor leads to a reduction in ϕ(η), i. e., mass diffusion into the boundary layer is suppressed. Figure 18 displays that as the mass transfer Biot number $\gamma_{2}\;$ increases, there is an enhancement in the magnitudes of the concentration field ϕ(η). This key parameter appears in the relevant boundary conditions for the concentration, i.e., $\phi =-\gamma_{2}\left(1-\phi \left(\eta \right)\right)$. Greater values of $\gamma_{2}\;$ significantly boost the concentration of the nanoparticles, but sequentially decrease the gradient of concentration at the wall. In Figure 19, it is evident that with higher values of the destructive parameter γ>0, the concentration profile of Oldroyd-B nanofluid exhibits a diminishing trend. With larger values of the generative parameter γ<0, the concentration of nanoparticles increases. Practically, it is found that when the reaction rate in the fluid is increased, greater conversion of the original nanoparticle species is induced in the presence of a chemical reaction, and thus the concentration of the original nanoparticles reduces. On the other hand, when the reaction rate in the liquid is decreased, fewer original species are converted and the nanoparticle concentration values are increased.

The numerical findings for NuxRex−12 and ShxRex−12 for diverse values of $\beta_{1}$, $\beta_{2}$, $\theta_{R}$, γ, $N_{t}$ and $N_{b}$ are disclosed in Table 4. Here, it can be seen that the heat and mass transfer rates are suppressed as the values of the relaxation $\beta_{1}$ and retardation $\beta_{2}$ fluid parameters increase, respectively. Since the stronger values of $\beta_{1}$ and $\beta_{2}$ upsurge the relaxation and retardation time parameters of the Oldroyd-B liquid, a reduction in the rate of heat and mass transfer at the wall is induced. The heat and mass transfer rates are boosted for greater values of $\theta_{R}$. When the liquid temperature $T_{f}$ is greater than the ambient temperature $T_{\infty }$, the thermal conductivity of the nanofluid increases as $\theta_{R}$ increases. Heat transfer to the wall is therefore enhanced. Moreover, as the chemical reaction variable γ increases, the concentration and temperature magnitudes in the boundary layer are suppressed, but the transport of nanoparticles and heat to the wall is elevated. NuxRex−12 and ShxRex−12 are therefore increased. It is also apparent that the heat and mass transfer rates are suppressed as the Brownian and thermophoretic parameter values increase. This is consistent with the results described earlier wherein temperatures and nanoparticle concentrations were boosted with these parameters. This leads to a depletion in the migration of heat and nanoparticle species to the wall away from the boundary layer and explains the plummet in local Nusselt and Sherwood number functions with higher Brownian motion and thermophoresis parameters. Again, the prescription of appropriate nanoparticles is critical for developing the desired nano-coating properties for delicate micro-machining applications since heat, mass and momentum characteristics are strongly influenced by nanoparticle mass diffusivity, nanoparticle thermal conductivity and the nanofluid viscosity, which is a function of nanoparticle concentrations.

## 7. Closing Remarks 

In the current article, we developed a novel mathematical model for the nonlinear mixed convection flow of Oldroyd-B type nanoliquids subjected to a chemical reaction from the stretching surface. This study was inspired by more rigorously analyzing the stagnation flow coating regimes in micro-machine and robotic manufacturing. The impacts of thermophoresis and Brownian motion features induced by nanoparticles were investigated using Buongiorno’s nanoscale model. High-temperature heat transfer was evaluated considering a nonlinear version of thermal radiation. The consistent momentum, energy and nanoparticle volume fraction equations were altered into a set of nonlinear ordinary differential ones and then solved analytically utilizing the homotopy algorithm. Comparisons of the HAM results with earlier studies were included. The assessment of convergent series solutions are also presented. The main findings emerging from the elaborated analysis are summarized below:

Influences of fluid relaxation and retardation time parameters are qualitatively opposite for the velocity profile of the Oldroyd-B nanoliquid (coating).Increments in the mixed convection parameter improves the velocity profile.Temperature and nanoparticle concentrations show an increasing trend for higher values of the thermophoresis parameter.A rise in the temperature ratio parameter leads to higher temperature magnitudes.Stronger values of thermal and solutal Biot numbers cause an augmentation in fluid temperature and the concentration of nanoparticles.An enhancement in Brownian motion transfers kinetic energy to heat energy, which increases the temperature of the nanofluid and elevates the thickness of the thermal boundary layer.The current Oldroyd-B viscoelastic nanofluid model reduces to the classical viscous fluid case when $\beta {\;}_{1}\;=\;\;\beta {\;}_{2}\;=\;\;0$. Important thermal, hydrodynamic and species transport characteristics can be captured with the Oldroyd-B viscoelastic formulation presented, which are not possible with classical Newtonian or simpler non-Newtonian models. The results presented are more realistic for actual nano-polymeric coating processes in robotics, sensor and micromachine surface deposition technologies.

The current study provides a good platform for further research into high temperature nano-polymeric coating flow processes. However, this study has neglected certain aspects including the consideration of variable sheet thicknesses, activation energy, Joule heating, entropy generation, Cattaneo–Christov heat–mass fluxes and bio-convective flows with gyrotactic microorganisms. These may be explored imminently.

## Figures and Tables

**Figure 1 micromachines-13-02196-f001:**
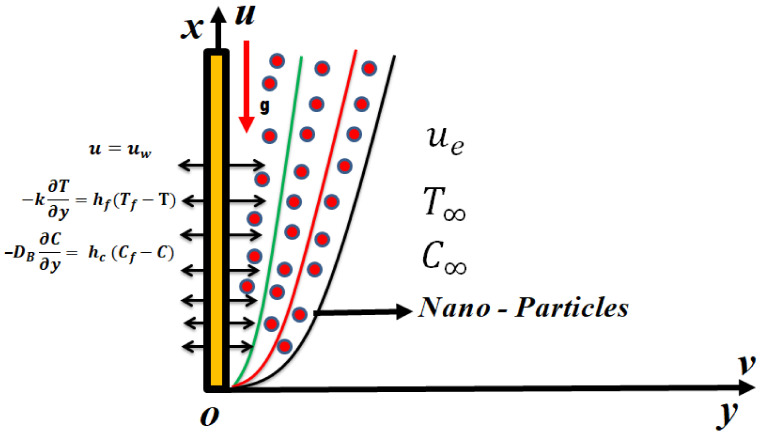
Flow model of Oldroyd-B nanoliquid.

**Figure 2 micromachines-13-02196-f002:**
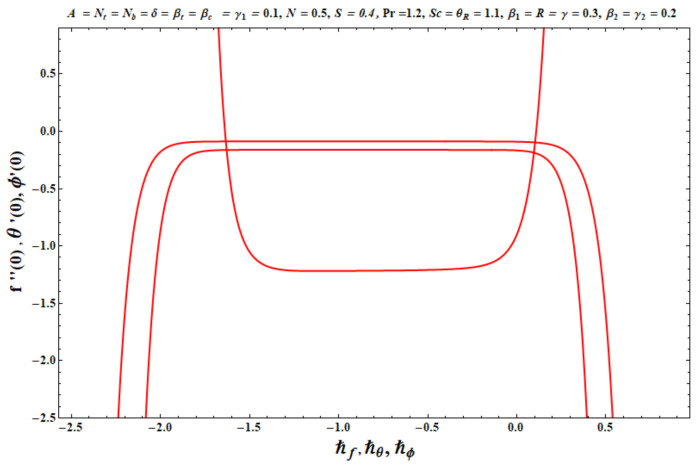
The ℏ
-curves against $f^{\prime \prime }\left(0\right),$ $\theta^{\prime }\left(0\right)\;\mathrm{and}\;\phi ’\left(0\right)$.

**Figure 3 micromachines-13-02196-f003:**
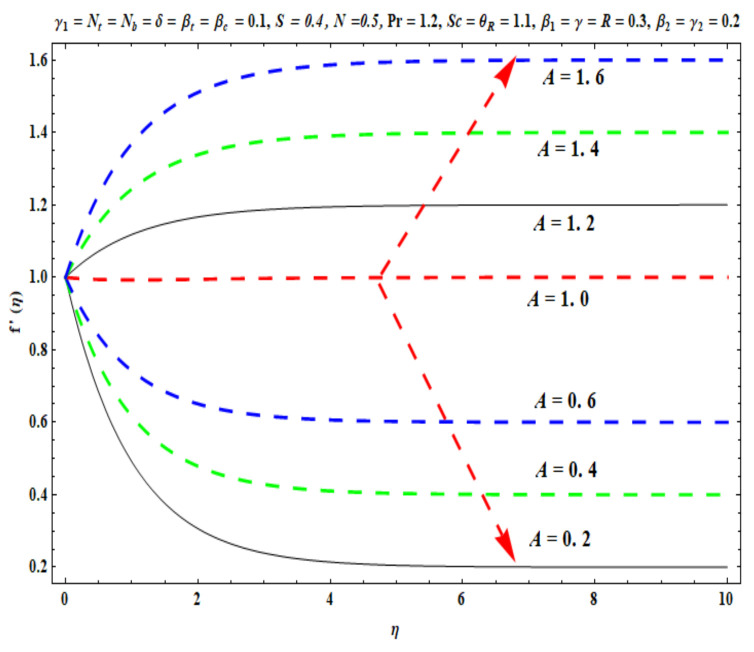
Against A .

**Figure 4 micromachines-13-02196-f004:**
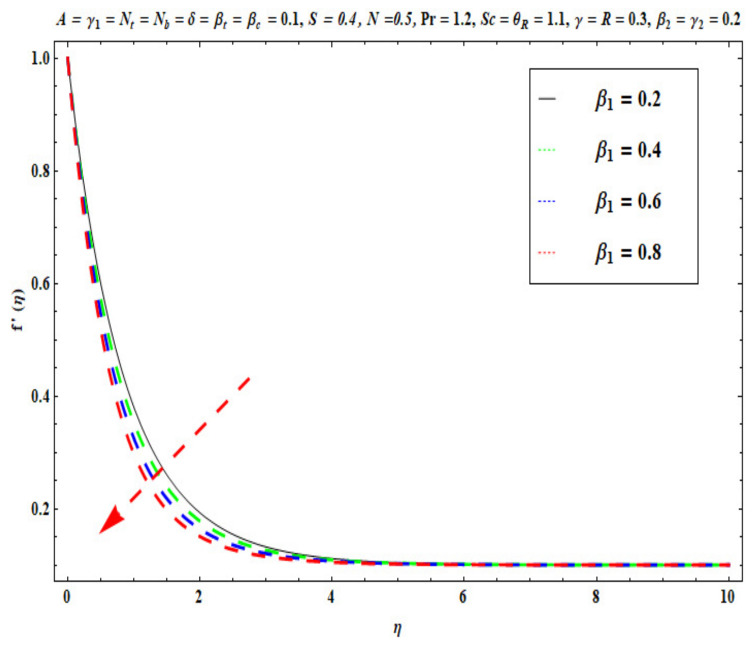
Against $\beta_{1}$.

**Figure 5 micromachines-13-02196-f005:**
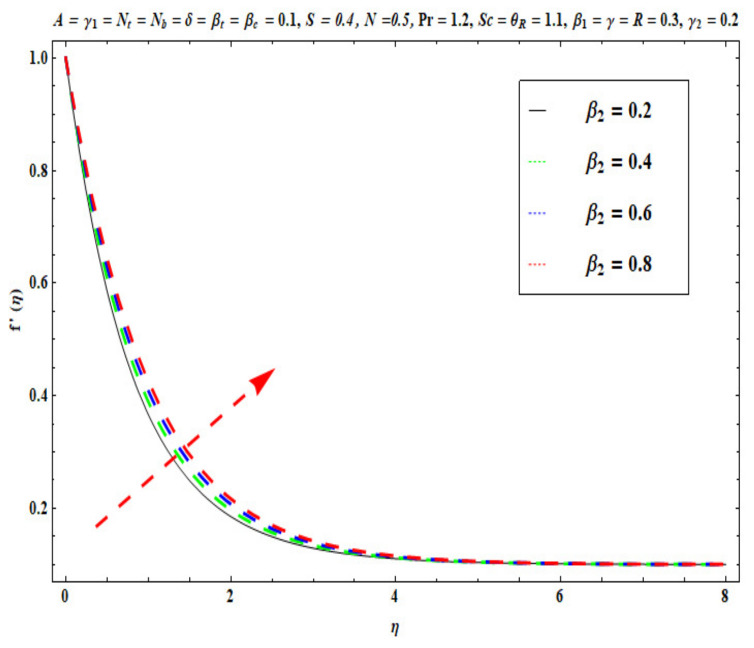
Against $\beta_{2}\;$.

**Figure 6 micromachines-13-02196-f006:**
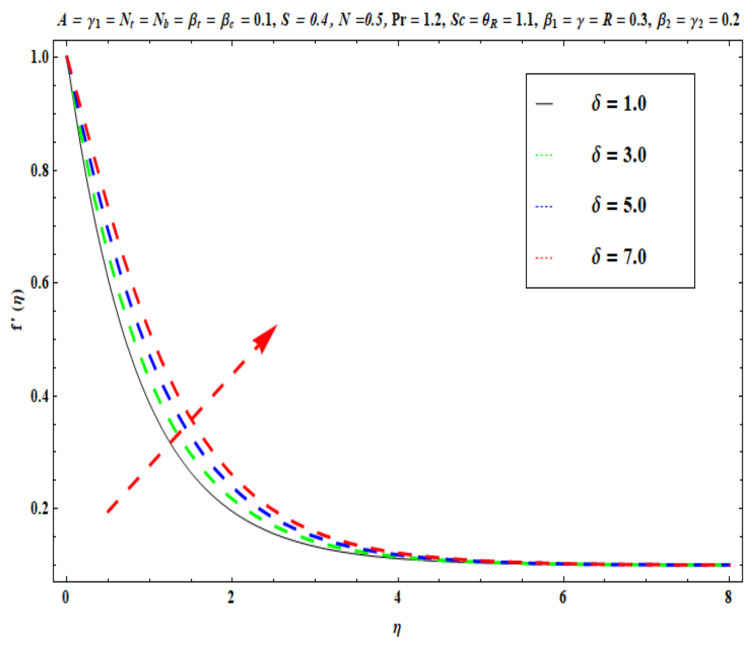
Against δ .

**Figure 7 micromachines-13-02196-f007:**
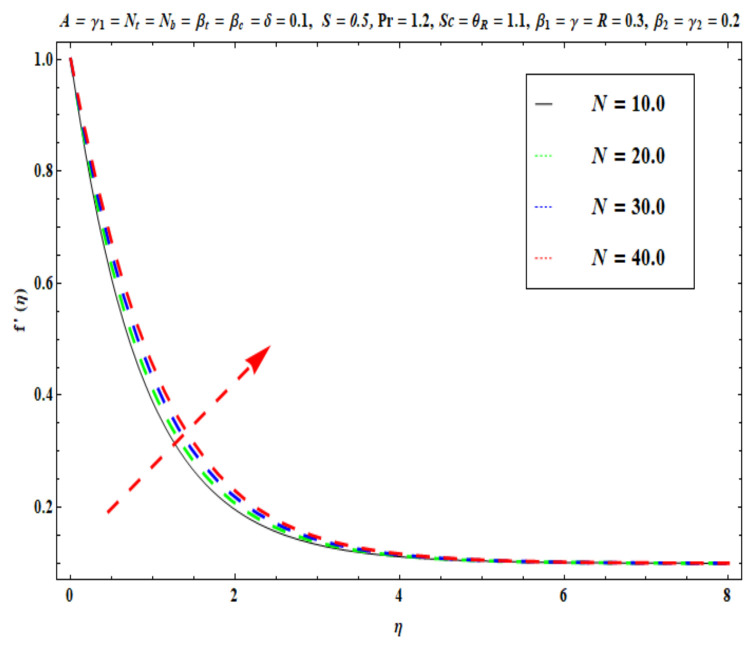
Against N .

**Figure 8 micromachines-13-02196-f008:**
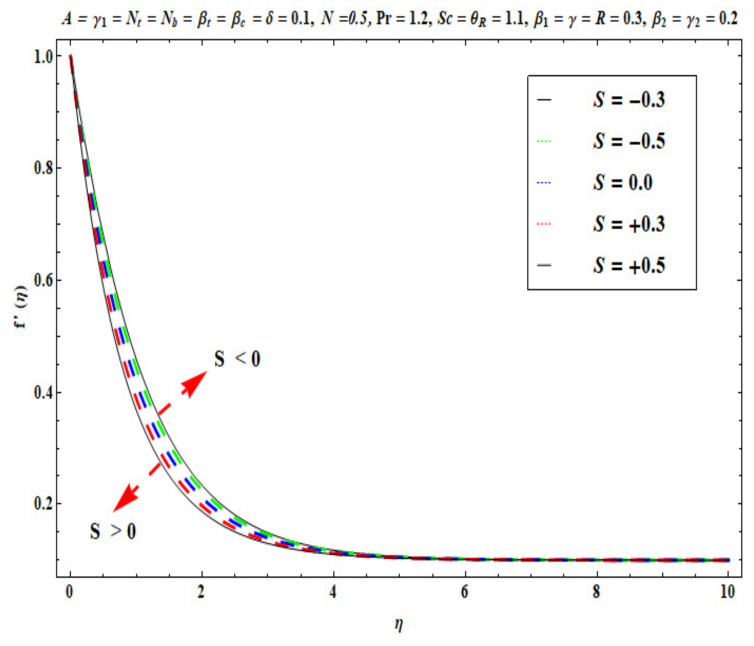
Against S .

**Figure 9 micromachines-13-02196-f009:**
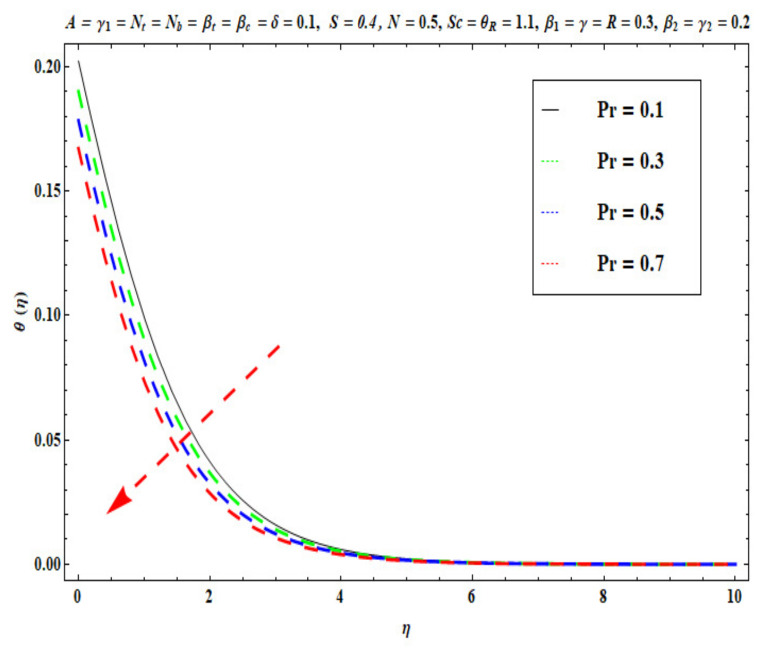
Against Pr .

**Figure 10 micromachines-13-02196-f010:**
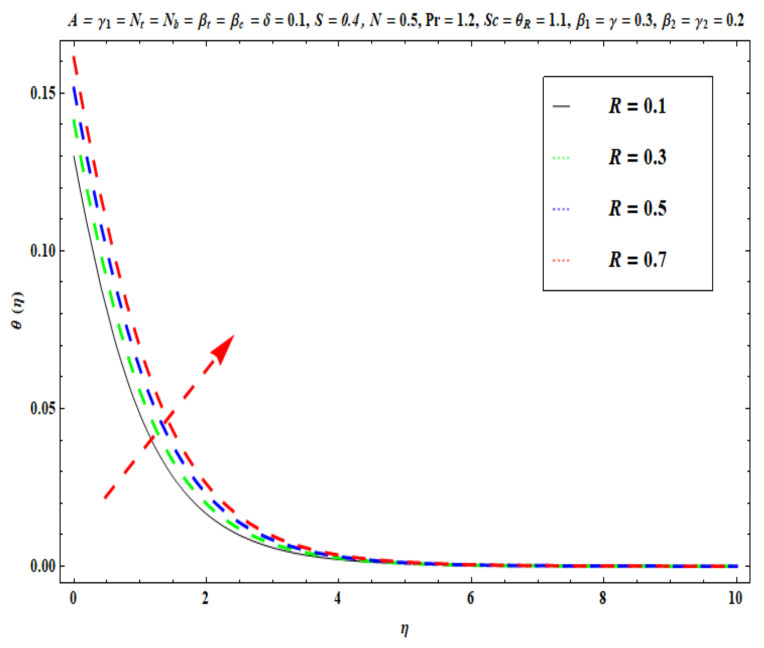
Against R .

**Figure 11 micromachines-13-02196-f011:**
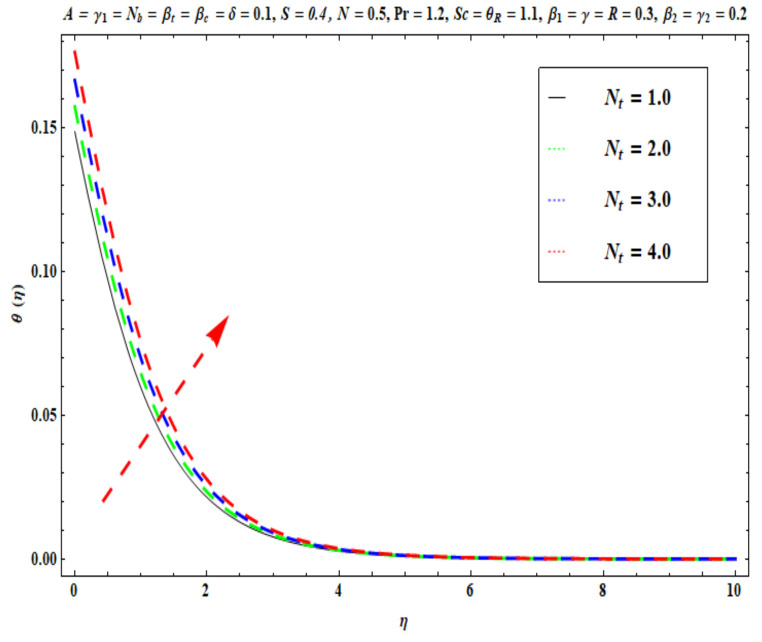
Against $N_{t}\;$.

**Figure 12 micromachines-13-02196-f012:**
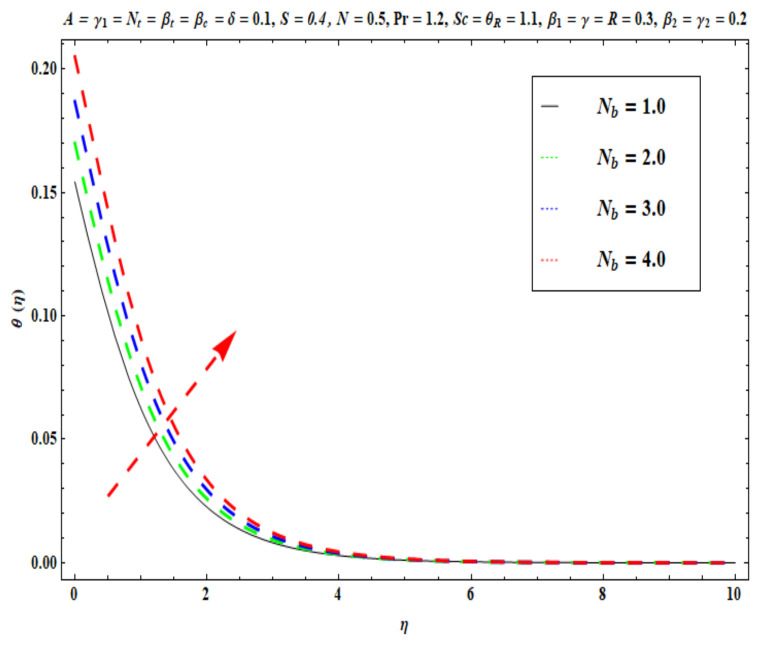
Against $N_{b}\;$.

**Figure 13 micromachines-13-02196-f013:**
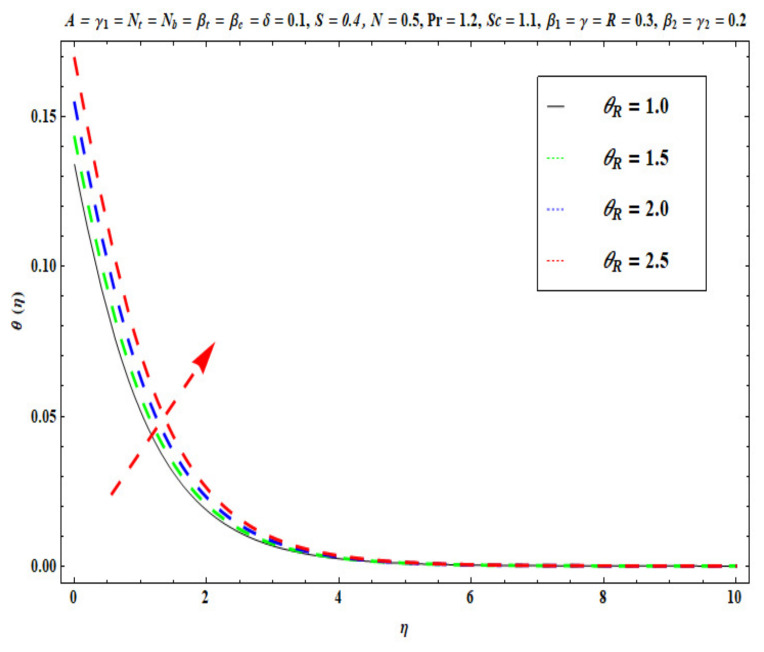
Against $\theta_{R}\;$.

**Figure 14 micromachines-13-02196-f014:**
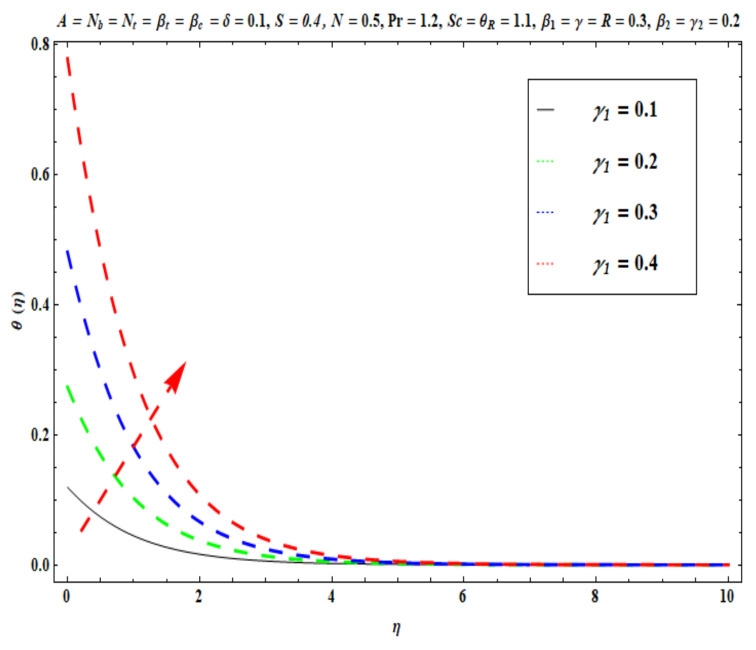
Against $\gamma_{1}\;$.

**Figure 15 micromachines-13-02196-f015:**
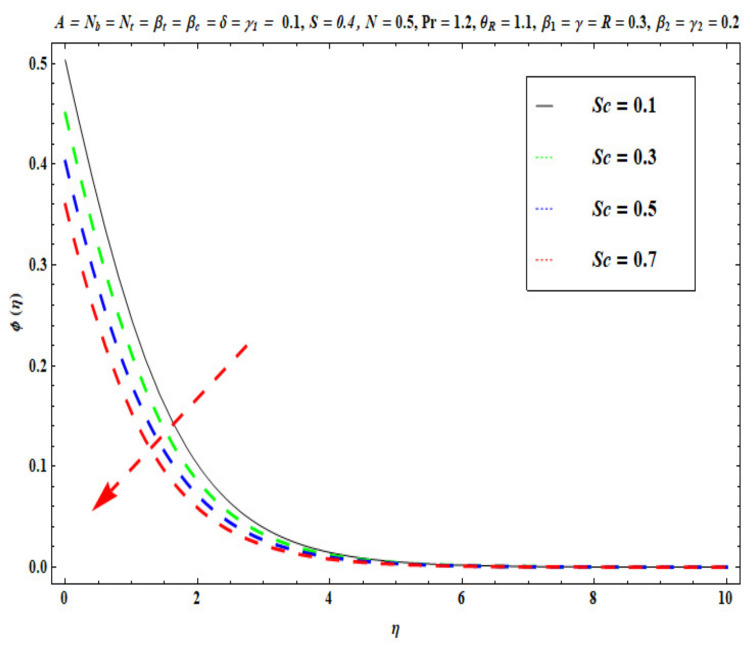
Against Sc .

**Figure 16 micromachines-13-02196-f016:**
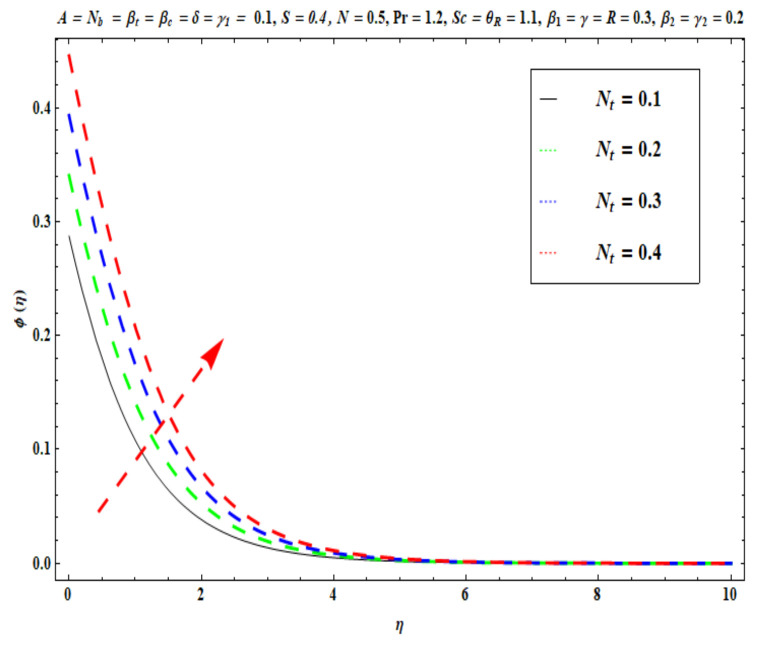
Against $N_{t}\;$.

**Figure 17 micromachines-13-02196-f017:**
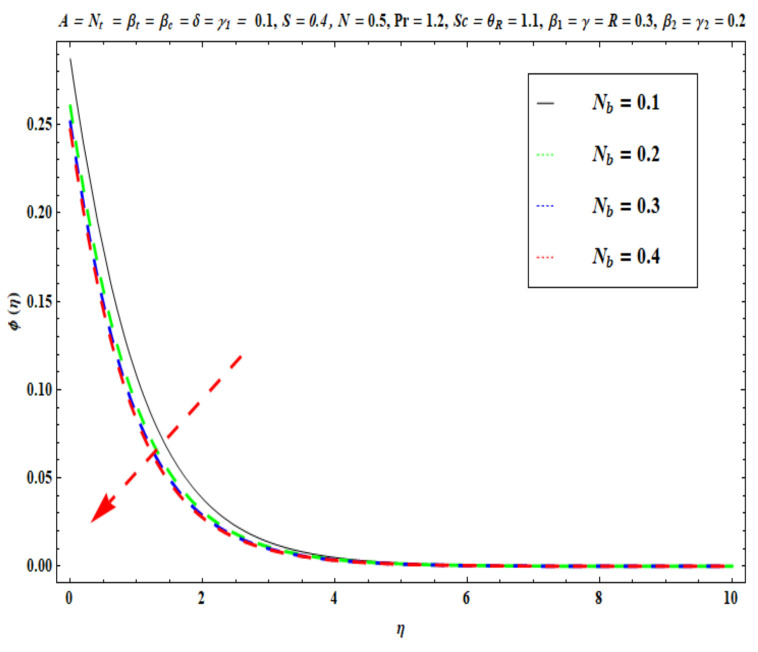
Against $N_{b}\;$.

**Figure 18 micromachines-13-02196-f018:**
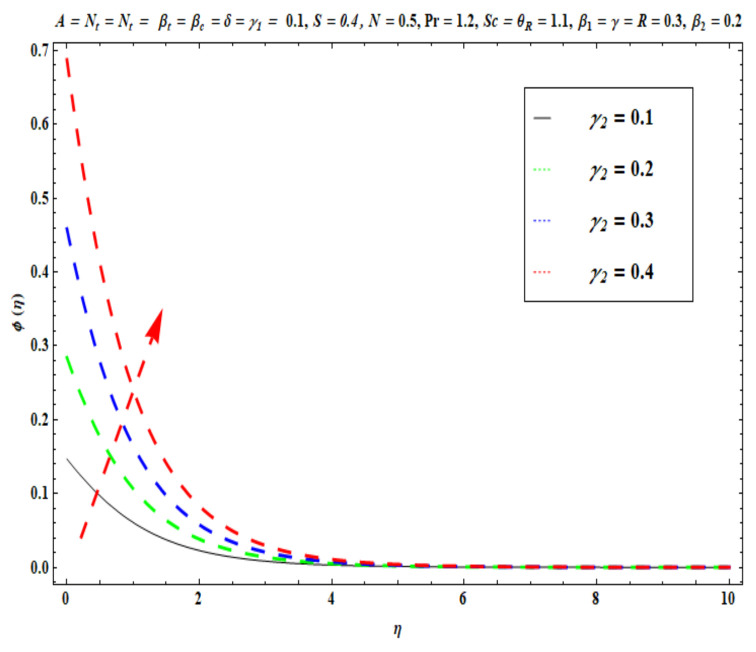
Against $\gamma_{2}\;$.

**Figure 19 micromachines-13-02196-f019:**
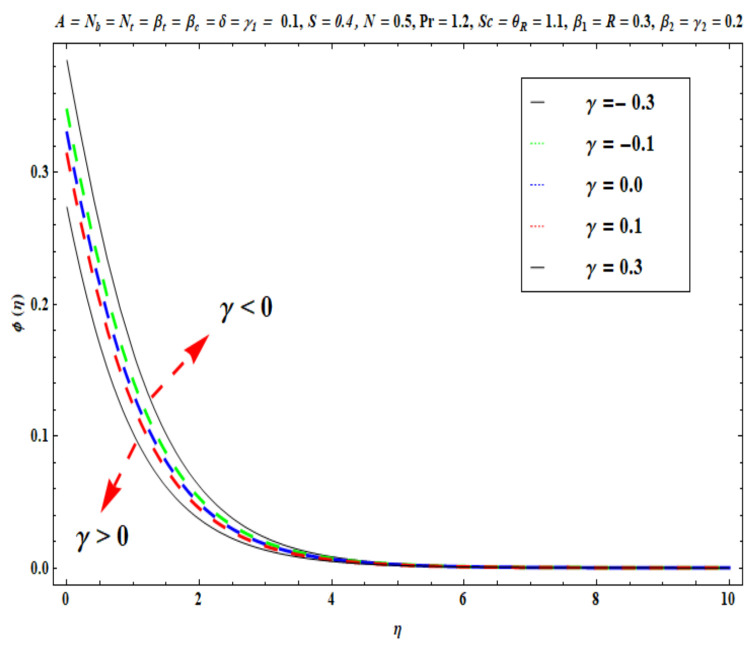
Against γ .

**Table 1 micromachines-13-02196-t001:** Assessment of convergence series solutions for the various order of approximations when $\gamma_{1}=\;\;A=\;\;\delta =\;\;N_{t}=\;\;N_{b}\;=\;\;\beta_{t}\;\;=\;\;\beta_{c}=0.1,\;\;N=\;0.5,\;\;S\;\;=\;0.4,\;\;\mathrm{Pr}=\;1.2,\;\;\theta_{R}=Sc\;\;=1.1,\;\;R\;\;=\;\;\beta_{1}\;=\;\;$
$\gamma \;\;=\;\;0.3\;\;\mathrm{and}\;\beta_{2}\;=\;\;\gamma_{2}\;=0.2$.

**Order of Approximations**	$-f^{\prime \prime }\left(0\right)$	$-\theta^{\prime }\left(0\right)$	$-\phi^{\prime }\left(0\right)$
1	1.0659	0.08967	0.1630
5	1.1889	0.08879	0.1625
10	1.2078	0.08868	0.1622
15	1.2136	0.08866	0.1621
20	1.2164	0.08867	0.1620
25	1.2187	0.08867	0.1620
30	1.2187	0.08867	0.1620
35	1.2187	0.08867	0.1620

**Table 2 micromachines-13-02196-t002:** Comparison of the results of $f^{\prime \prime }\left(0\right)$ for $\beta_{1}$ when A = δ =0.

$\beta_{1}$	Ref. [31]	Ref. [32]	Present Study
0.0	−1.000000	−1.000000	−1.000000
0.2	−1.051889	−1.051838	−1.051901
0.4	−1.1019032	−1.101842	−1.101841
0.6	−1.1501373	−1.150231	−1.150231
0.8	−1.1967113	−1.197532	−1.196711
1.2	−1.2853632	−1.285317	−1.285316

**Table 3 micromachines-13-02196-t003:** Comparison of the results of $f^{\prime \prime }\left(0\right)$ for $\beta_{2}$ when A = δ =0 with fixed $\beta_{1}\;=\;\;0.4$.

$\beta_{2}$	Ref. [49]	Ref. [50]	Present Study
0.0	1.10190	1.10193	1.10193
0.2	1.00498	1.00493	1.00493
0.4	0.92986	0.92975	0.92975
0.6	0.86942	0.86654	0.86654
0.8	0.81943	0.81932	0.81932
1.0	0.77718	0.77681	0.77681

**Table 4 micromachines-13-02196-t004:** Numerical values of NuxRex−12 and ShxRex−12 considering A<1.

$\beta_{1}$	$\beta_{2}$	$\theta_{R}$	γ	$N_{t}$	$N_{b}$	NuxRex−12	ShxRex−12
0.1	0.1	1.1	0.3	0.1	0.1	0.1360	0.1623
0.2						0.1359	0.1621
0.3						0.1358	0.1619
0.1	0.2					0.1361	0.1624
	0.3					0.1362	0.1625
	0.4					0.1363	0.1626
	0.1	1.2				0.1499	0.1624
		1.3				0.1662	0.1625
		1.4				0.1852	0.1626
		1.1	0.4			0.1360	0.1642
			0.5			0.1361	0.1659
			0.6			0.1362	0.1674
			0.3	0.2		0.1359	0.1546
				0.3		0.1358	0.1469
				0.4		0.1357	0.1422
				0.1	0.2	0.1359	0.1564
					0.3	0.1358	0.1453
					0.4	0.1357	0.1413

## Data Availability

Data regarding current research is mentioned in the manuscript.

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
