# Peer review of "Analysis of Nonlinear Convection–Radiation in Chemically Reactive Oldroyd-B Nanoliquid Configured by a Stretching Surface with Robin Conditions: Applications in Nano-Coating Manufacturing"

_micromachines, 2022, doi:10.3390/mi13122196_

Round 1

Reviewer 1 Report

 The authors in this communication models the nonlinear thermally radiative Oldroyd-B nanoliquid stagnant-point flow configured by heated vertical surface. Stretching permeable surface satisfying Robin conditions engenders the flow. The nanoliquid analysis is based on Buongiorno's model which elaborates Brownian movement and thermophoretic attributes. Nonlinear version of buoyancy forces together with nonlinear thermal radiation is introduced. Chemical reaction (constructive and destructive) is considered. Similarity approach assists in obtaining ordinary differential systems from partial ones. Analytical soutions are achieved employing homotopy analysis scheme. This paper is supported for publication. Only the following requires attention of authors.

1.     How the equation (2) has been derived? It should include a brief discussion or at least support of any previous study, so that readers can understand it quickly.

2.     Applications of boundary-layer should be incorporated in the revised manuscript.

3.     Why Oldroyd-B fluid is considered?

4.     The scientists and researchers working in boundary-layer direction should be encouraged by mentioning the future directions of present work so that they can extend their work in this direction.

5.     What physical situation is being modeled by the flow?

6.     Pr=1.2 for most of the calculations. The authors should clearly indicate for which matter it stands. In fig. 9 it varies from 0.1 to 0.7. Since the fluid taken is non-Newtonian type it has some definite value.

7.     As in the case of generalized non-Newtonian fluid the energy equation is modified with the modified viscosity parameter (Bird et al 2000), but here energy equation is used as in the case of Newtonian fluid. Authors should clearly justify that it can also be used for Oldroyd-B fluid model.

8.     Explain, why homotopic analysis method is better. Because this type of works already done.

9.     The discussions of results are not sufficient and the author should add extra explanation.

10.  How the results for viscous fluid can be achieved from present study?

11.  I suggest you cite few references from 2022.

Author Response

Reviewer # 1

  1. 1: How the equation (2) has been derived? It should include a brief discussion or at least support of any previous study, so that readers can understand it quickly.

Reply: A suitable reference in support of Eq. (2) and other governing expressions is included now in the revised manuscript (for detail see at page 3).

  1. 2: Applications of boundary-layer should be incorporated in the revised manuscript.

Reply: It is done (for detail see the first paragraph of introduction at page 1).

  1. 3: Why Oldroyd-B fluid is considered?

Reply: The reason of choosing Oldroyd-B fluid is included in the introduction (for detail see at page 2).

  1. 4: The scientists and researchers working in boundary-layer direction should be encouraged by mentioning the future directions of present work so that they can extend their work in this direction.

Reply: It is done (for detail see the last point of conclusions at page 15).

  1. 5: What physical situation is being modeled by the flow?

Reply: Physical situation of considered model is explained through Fig. 1.

  1. 6: Pr=1.2 for most of the calculations. The authors should clearly indicate for which matter it stands. In fig. 9 it varies from 0.1 to 0.7. Since the fluid taken is non-Newtonian type it has some definite value.

Reply: We have discussed the Maxwell fluid model in the present problem. This model has many examples in the real life but we have analyzed the whole Maxwell fluid model, not any specific example. Therefore, we have selected different values of Prandtl instead of any specific value. If we use any specific example, then we have to used specific value of Prandtl number corresponding to the specific material and its variation is not possible in that case.

  1. 7: As in the case of generalized non-Newtonian fluid the energy equation is modified with the modified viscosity parameter (Bird et al 2000), but here energy equation is used as in the case of Newtonian fluid. Authors should clearly justify that it can also be used for Oldroyd-B fluid model.

Reply: Modification in the energy equation for various fluid models is associated with the viscous dissipation term which differs through extra stress tensor for a particular fluid. But we have neglected the viscous dissipation effects in this problem. Therefore, energy equation is same as for Newtonian and non-Newtonian fluids in the absence of viscous dissipation. Physically such consideration is reliable for slow speed of fluid. This fact is already utilized abundantly in the existing literature. However, it should be noted that various parameters such as thermal conductivity k, specific heat Cp and density ρ are of non-Newtonian fluids which have different values from Newtonian fluids.

  1. 8: Explain, why homotopic analysis method is better. Because this type of works already done.

Reply: We adopted the homotopy analysis method (HAM) in view of the following aspects.

  • The homotopy method does not comprise any large/small parameters.
  • The convergence of established expressions can be justified easily.
  • HAM delivers great liberty to choose the linear operators and base functions.
  1. 9: The discussions of results are not sufficient and the author should add extra explanation.

Reply: We have significantly extended the discussion and included implications in terms of the application area which is non-Newtonian nanofluid coating dyamics for micromachines. See the revised text:

Please see the new text: 

“Figs. 3-9 disclose the impact of ,   ,    and  on velocity . Fig. 3. demonstrates that an increment in the velocity ratio factor ,  indicates an improvement with respect to η in the boundary regime, that is, the fluid motion is accelerated on the stretchy surface. When  free stream velocity is stronger than the stretching velocity of the surface, this builds a momentum upsurge in the flow domain through exterior free stream which exhibits high acceleration for all horizontal coordinate values, η. Consequently, the thickness of the momentum boundary layer of the stretching surface is therefore decreased. This has an influence on the quality control of manufactured coatings. However, when  the stretching velocity of the sheet is higher than the exterior velocity of free stream and the reverse impact is calculated.  i.e., velocity of Oldroyd-B fluid  is decreased and thickness of momentum boundary layer on the surface increases. If  both the external and the stretchy velocity of the Oldroyd-B fluid are identical. This scenario is a natural intermediate between cases where A > 1 and A< 1. It is evident that higher stretching sheet prevents the development of momentum; however, a stronger external velocity of corresponding Oldroyd-B fluid is produced. The influence of Deborah numbers on the velocity of Oldroyd-B fluid  that is in terms of (  relaxation time quantity) and (  retardation time quantity) are portrayed in Fig. 4 and 5. respectively. Here in Fig. 4, it is clear that as (relaxation time fluid)  rises, the Oldroyd-B fluid velocity gradually declines. In fact, is mathematically expressed as ‘‘the ratio of the observational timescale to the time scale of the material reaction”. We can examine the three different scenarios to assess the polymeric behavior of materials. (i) When  for viscoelastic substance, (ii) when  for entirely viscous material, (iii) when  for elastic material in nature. Higher values of  lead to a lower relaxation relative to the characteristic timescale. This means that fluid react in a similar way to solid materials. Fig. 5 displays the impact of retardation time  parameter on Oldroyd-B fluid velocity . As anticipated,  upsurges when  is increased. Physically, retardation time augments for increasing  owing to which  upsurges. Fig. 6 reveals the features of mixed convection variable  on . Clearly  increases when the mixed convection variable is elevated. Since thermal buoyancy forces exceed the viscous forces with higher values of , this  intensifies the flow. Fig. 7 unveils the impact of buoyancy ratio variable on . It can be seen that a massive boost is induced in the linear velocity across the domain with stronger values of . Note that  signifies that the concentration buoyancy force   of nanoparticles is much stronger than the temperature buoyancy force . Consequently, the hydrodynamic boundary layer thickness of Oldroyd-B fluid increases. Fig. 8 depicts the characteristics of wall suction/injection on . It is analyzed that the impetus of linear velocity is increased with amplification in the injection parameter . On the other hand, it is decreased for greater values of wall suction factor . It is evident that injection of nanofluid augments momentum development in the boundary layer regime and, consequently, velocity of Oldroyd-B nanofluid increases. This effect is also known as blowing in manufacturing processes. The reverse situation is noticed for suction which causes the boundary layer to adhere more strongly to the wall and decelerates flow- this increases momentum boundary layer thickness. 

6.2. Temperature Profile

Figs.9–14 display the effect of          and  on . Fig. 9 depicts the  curves subjected to Prandtl number  values. This Fig. exhibits that the nanofluid temperature decreases for larger Prandtl number. Prandtl number ( ) communicates the rate of momentum to thermal diffusion. Liquid conductivity retards for larger  The heat transferred via conduction of molecules is subsequently repressed which expresses a decay in  and a diminution in thickness of thermal boundary-layer. Chilling of stretched coating system is thus attained with increasing  while heating is witnessed with lower . Fig. 10 illustrates the curves of  under the impact of . It is witnessed that when is increased, is obviously enhanced. Actually, greater heat (thermal energy) is produced in the working liquid during the radiation process (i.e., for higher R values), causing the temperature of the nanofluid in the boundary layer regime to increase. Fig. 11 divulges the consequence of Buongiorno's model parameter (nanoscale thermophoresis ) on thermal distribution  of Oldroyd-B fluid. Stronger movement of nanoparticles in the stagnation boundary layer flow domain is encouraged by the thermophoretic body force; nanoparticles are mobilized  from the hotter region to the cooler one and therefore higher thermal transmission arises in the flow field. Hence, intensifying magnitude of nanoscale thermophoresis parameter  produces substantial thermal diffusion enhancement and boosts temperature and thermal boundary layer thickness. The influence of the Brownian motion factor  on the temperature  of the Oldroyd-B nanofluid are presented in Fig. 12. There is an accentuation in nanofluid temperature  and also thermal boundary-layer thickness with increment in . In fact, the random movement of the fluid particles is increased since nanoparticle diameters are reduced with higher Brownian motion parameter and this generates intensification in ballistic collisions which produces extra heat in the regime. Micro-convection around the nanoparticles is also enhanced with greater Brownian motion effects. All these factors contribute to marked thermal enhancement.The effect of temperature ratio variable  on temperature distribution  is illustrated in Fig. 13. Physically, temperature appears to rise significantly as  increases. Higher values of  implies an elevation in wall temperature which makes the depth of thermal penetration deeper into the boundary layer. Heat transfer into the flow from the wall is therefore encouraged. Also, when the liquid temperature  exceeds the ambient temperature  (in energy equation) this creates a larger temperature differential across the boundary layer which intensifies thermal diffusion from the wall to the free stream and manifests in a boost in  rises. Fig. 14 shows that with larger values of thermal Biot number , there is an improvement in the temperature field . The condition  suggests the configuration of isoflux at the wall, while  symbolizes the configuration of an isothermal wall. In addition, this parameter is featured in the prescribed boundary condition   (from Eqn. (11)) which determines the intensity of Biot number. Therefore, stronger values of Biot number  corresponds to an amplification in thermal convection over the stretching sheet which enhances the nanofluid temperature. The inclusion of this complex convective wall boundary condition provides a more realistic estimation of manufacturing conditions than conventional boundary conditions.  

6.3. Concentration Profile

Figs.15–19 show the impact of   ,    and on . Fig. 15 reveals that with higher values of  the nanoparticle concentration  and thickness of solutal boundary-layer reduces. Actually, Schmidt number is the ratio of momentum to the nanoparticle molecular diffusivity which implies that when rises, mass diffusivity reduces and there is a depletion in concentration . The judicious selection of nanoparticles to embed in the coating regime is therefore critical in achieving bespoke mass transfer characteristics in fabrication processes. Figs. 16 and 17 portray the evolution in concentration   of nanoparticles with different values of thermophoresis  and Brownian   diffusion parameters. When  increases, an increasing trend is seen via higher chaotic movement associated with higher , the particle collision is boosted and mass diffusion is assisted. As a result, the Oldroyd-B nanofluid concentration upsurges. The reverse trend is noted for the influence of thermophoresis parameter  on the surface concentration . Physically, the movement of nanoparticles from the wall to the interior boundary layer region is impeded by an elevation in the thermophoretic force. Therefore, this factor leads to reduction of  i. e. mass diffusion into the boundary layer is suppressed. Fig. 18 displays that with increasing values of mass transfer Biot number , there is an enhancement in magnitudes of the concentration field . This key parameter appears in the relevant boundary conditions of concentration.  i.e.  . Greater values of  significantly boost the concentration of nanoparticles but decrease the gradient of concentration at the wall sequentially. In Fig. 19, it is evident that higher values of destructive parameter , the concentration profile of Oldroyd-B nanofluid exhibits a diminishing trend. With larger values of generative parameter , the concentration of nanoparticles increases.  Practically, it is found that when the reaction rate in the fluid is increased, then greater conversion of the original nanoparticle species is induced in the presence of chemical reaction and thus the concentration of the original nanoparticles reduces. On the other hand, when the reaction rate in the liquid is decreased, then less original species is converted and nanoparticle concentration values are increased.

The numerical findings for and  for diverse values of , , , ,   and  are disclosed in Table. 4. Here it is examined that, the heat and mass transfer rates suppressed for the growing values of relaxation  and retardation  fluid parameters, respectively. Since the stronger values of  and  upsurge the relaxation and retardation times parameters of Oldroyd-B liquid, this induces a reduction in rate of heat and mass transfer at the wall. The heat and mass transfer rates are boosted for greater values of . When the liquid temperature  is greater than the ambient temperature  this effectively increases the thermal conductivity of nanofluid, as  increases. Heat transfer to the wall is therefore enhanced. Moreover, the chemical reaction variable  increasing suppresses concentration and temperature magnitudes in the boundary layer but elevates transport of nanoparticles and heat to the wall.  and  are therefore increased. It is also apparent that heat and mass transfer rates are suppressed with increasing values of Brownian and thermophoretic parameters. This is consistent with the results described earlier wherein temperatures and nanoparticle concentrations were boosted with these parameters. This leads to a depletion in the migration of heat and nanoparticle species to the wall away from the boundary layer and explains the plummet in local Nusselt and Sherwood number functions with higher Brownian motion and thermophoresis parameters. Again the prescription of appropriate nanoparticles is critical in developing the desired nano-coating properties for delicate micro-machining applications since heat, mass and momentum characteristics are strongly influenced by for example nanoparticle mass diffusivity, nanoparticle thermal conductivity and also the nanofluid viscosity which is a function of the nanoparticle concentration.”

  1. 10: How the results for viscous fluid can be achieved from present study?

Reply: The results of viscous fluid can be achieved in the absence of material parameter. The current Oldroyd-B viscoelastic nanofluid model reduces to the classical viscous fluid case when . (see text)

  1. 11: I suggest you cite few references from 2022.

Reply: It is done Please note we have also added 5 new references for applications to elucidate the actual applications of the regime studied. See:

[51] Haupt, K.; Mosbach, K. Molecularly imprinted polymers and their use in biomimetic sensors. Chem. Rev.  100, 2495–2504 (2000). 

[52] Stolov, A.A.; Wrubel, J.A.; Simoff, D.A. Thermal stability of specialty optical fiber coatings. J. Therm. Anal. Calorim. 2016, 124, 1411–1423 (2016).

[53] 15. Rivero, P.J.; Goicoechea, J.; Arregui, F.J. Optical fiber sensors based on polymeric sensitive coatings. Polymers, 10, 280 (2018).

[54] K. M. Sivaraman et al., "Functional polypyrrole coatings for wirelessly controlled magnetic microrobots," 2013 IEEE Point-of-Care Healthcare Technologies (PHT), 2013, 22-25 (2013).

[55] J. Yang et al., Conformal surface-nanocoating strategy to boost high-performance film cathodes for flexible zinc-ion batteries as an amphibious soft robot, Energy Storage Materials, 46, 472-481 (2022).

Reviewer 2 Report

The paper needs careful major revision in the light of following queries before this work can be accepted for publication:

1.     Mathematical modeling is also not well presented. Every equation must be named and discussed. Model simplifications must be discussed and all variables must be defined.

2.     Nowadays we are not solving an equation to show that we are capable of solving it. We should also show that what is the benefit of solving an equation.

3.     Revise the introduction such that each paragraph shall present the meaning of a concept/keyword.

4.     Please discuss at the end of introduction, that why is it important from the aspect of APPLICATION and future experiments to investigate magneto-nanofluid mixed convective flow caused by a vertical permeable plate influenced by nonlinear thermal radiation? This can improve your impact on the field. Please also focus on connecting initial motivation with conclusions at the end. In this format the paper is more of a list of case studies, with parameter analysis and without deeper discussion on how these parameter changes are affecting the topic.

5.     Research questions are needed. Note that the results in this report are typical answers to unknown questions.

6.     It is necessary to discuss the justification for the applicability of such an approximation. The description of the conditions of applicability of the initial model (1) - (6) is the most important component of the proposed study.

7.     There is no momentum equation in y direction?

8.     We must have a justification for similarity transformations (7), since self-similar solutions greatly simplify the flow structure. Such solutions may not describe the actual physical system.

9.     There is no remarkable finding in the investigation; all those are usual and already obtained previously. In addition, in the “Results and discussion” section the many results are just described (which can be observed by anyone from the figures) without proper physical reasoning

10.  Please discuss at the end of introduction, that why is it important from the aspect of APPLICATION and future experiments to investigate the current study.

11.  The units of the parameters can be included.

  1. In the last paragraph of the introduction section authors are requested to mention all the additional parameters considered in this work. The ones which are not already reported in the literature. Justification must be given why these additional parameters are important.

Author Response

Reviewer # 2

  1. 1: Mathematical modeling is also not well presented. Every equation must be named and discussed. Model simplifications must be discussed and all variables must be defined.

Reply: The model is clearly described and all equations are correct. Justification is given in the introduction section for the modelling. Please read it carefully. If we justified every single aspect the paper would be too long! We have to be concise and expect readers to have some basic knowledge of fluid mechanics and nanofluidics and heat transfer!

See the revised text:

Consider the nonlinear mixed convective flow in the stagnation region of Oldroyd-B nanoliquid confined by a heated permeable vertical surface, as a model of manufacturing coating deposition of a nanopolymer. The rheological nature of many polymeric nanocoatings requires a robist non-Newtonian model which can simulate real effects including relaxation and retardation, for which the Oldroyd-B model is an excellent candidate. The nanoliquid model presented by Buongiorno [8] featuring Brownian diffusion and thermophoresis effects is utilized since it provides two-component (thermosolutal) framework for analysis. Nonlinear radiative aspect along with Robin conditions is considered to represent high temperature manufacturing conditions and complex wall conditions in coating deposition processes. Mass transfer effects are explored considering chemical reaction which is also common in nanomaterial polymeric coating synthesis. The physical model is visualized in Fig. 1. The governing expressions under the considered effects are [34]:

  1. 2: Nowadays we are not solving an equation to show that we are capable of solving it. We should also show that what is the benefit of solving an equation.

Reply: The whole purpose of modelling is to get a solution. The comment is very basic and we expect better from a reviewer for Micromachines. The journal is an applied journal, not an abstract mathematical journal. The reviewer should focus on the relevance of his/her comments and try to appreciate the article for its focus.

  1. 3: Revise the introduction such that each paragraph shall present the meaning of a concept/keyword.

Reply: This is not a helpful comment. The introduction is logical and it is organized based on the analysis to follow. There is no need to restructure it based on this request of the reviewer.

  1. 4: Please discuss at the end of introduction, that why is it important from the aspect of APPLICATION and future experiments to investigate magneto-nanofluid mixed convective flow caused by a vertical permeable plate influenced by nonlinear thermal radiation? This can improve your impact on the field. Please also focus on connecting initial motivation with conclusions at the end. In this format the paper is more of a list of case studies, with parameter analysis and without deeper discussion on how these parameter changes are affecting the topic.

Reply: We have added the following text in the abstract, at the end of the introduction and also integrated implications of the application (nanocoating of micromachine components) in the discussion:

See:

Abstract modified:

Motivated by emerging high-temperature manufacturing processes deploying nano-polymeric coatings, the present communication studies nonlinear thermally radiative Oldroyd-B viscoelastic nanoliquid stagnant-point flow from a heated vertical stretching permeable surface. Robin (mixed derivative) conditions are utilized to better represent coating fabrication conditions. The nanoliquid analysis is based on Buongiorno's two-component model which elaborates Brownian movement and thermophoretic attributes. Nonlinear buoyancy force and thermal radiation formulations are included. Chemical reaction (constructive and destructive) is also considered since coating synthesis often features reactive transport phenomena. Via a similarity approach, an ordinary differential equation model is derived from the primitive partial differential boundary value problem. Analytical solutions are achieved employing homotopy analysis scheme. The influence of emerging dimensionless quantities on transport characteristics is comprehensively elaborated with appropriate data. The obtained analytical outcomes are compared with available limiting studies and good correlation is achieved. The computations show that the velocity profile is diminished with increasing relaxation parameter whereas it is enhanced when retardation parameter is increased. Larger thermophoresis parameter induces temperature and concentration enhancement. The heat and mass transfer rates at the wall are increased with an increment in temperature ratio and first order chemical reaction parameters while contrary effects are observed for larger thermophoresis, fluid relaxation and Brownian motion parameters. The simulations find applications in stagnation nano-polymeric coating of micromachines, robotic components and sensors.

At end of introduction

The applications of the present non-Newtonian nanofluid stagnation model includes coating manufacturing processes for biomimetic sensors [51], optical fiber nanocoatings [52,53] and micro-robot surface protection [54, 55].

5 new references are included at the end of the reference section:

[51] Haupt, K.; Mosbach, K. Molecularly imprinted polymers and their use in biomimetic sensors. Chem. Rev.  100, 2495–2504 (2000). 

[52] Stolov, A.A.; Wrubel, J.A.; Simoff, D.A. Thermal stability of specialty optical fiber coatings. J. Therm. Anal. Calorim. 2016, 124, 1411–1423 (2016).

[53] 15. Rivero, P.J.; Goicoechea, J.; Arregui, F.J. Optical fiber sensors based on polymeric sensitive coatings. Polymers, 10, 280 (2018).

[54] K. M. Sivaraman et al., "Functional polypyrrole coatings for wirelessly controlled magnetic microrobots," 2013 IEEE Point-of-Care Healthcare Technologies (PHT), 2013, 22-25 (2013).

[55] J. Yang et al., Conformal surface-nanocoating strategy to boost high-performance film cathodes for flexible zinc-ion batteries as an amphibious soft robot, Energy Storage Materials, 46, 472-481 (2022).

  1. 5: Research questions are needed. Note that the results in this report are typical answers to unknown questions.

Reply: The research questions are implicit in the novelty. The inclusion of complex Robin conditions, non-Newtonian effects and other phenomena are analyzed collectively since we are investigating the combined influence of these aspects to present a more realistic simulation of actual nanocoating processes.

  1. 6: It is necessary to discuss the justification for the applicability of such an approximation. The description of the conditions of applicability of the initial model (1) - (6) is the most important component of the proposed study.

Reply: The approximations have been justified in the mathematical modelling section. Every model must make assumptions. No model can represent all phenomena. Boundary layer theory provides a solid foundation for the modelling since coating flow is examined. Robin boundary conditions represent more realistically convective effects at the wall. Nanofluid rheology requires a non-Newtonian model. High temperature invokes radiative flux. We have clarified this as succinctly as we can in the modelling section.

See the revised introduction to modelling section 2:

Consider the nonlinear mixed convective flow in the stagnation region of Oldroyd-B nanoliquid confined by a heated permeable vertical surface, as a model of manufacturing coating deposition of a nanopolymer. The rheological nature of many polymeric nanocoatings requires a robist non-Newtonian model which can simulate real effects including relaxation and retardation, for which the Oldroyd-B model is an excellent candidate. The nanoliquid model presented by Buongiorno [8] featuring Brownian diffusion and thermophoresis effects is utilized since it provides two-component (thermosolutal) framework for analysis. Nonlinear radiative aspect along with Robin conditions is considered to represent high temperature manufacturing conditions and complex wall conditions in coating deposition processes. Mass transfer effects are explored considering chemical reaction which is also common in nanomaterial polymeric coating synthesis. The physical model is visualized in Fig. 1. The governing expressions under the considered effects are [34]:

  1. 7: There is no momentum equation in y direction?

Reply: In boundary layer models, the whole purpose is to include only an x-direction momentum equation. Only when secondary flow is present e.g. in rotating systems, does one require a y-direction momentum equation. Please see the classical text by Schlichting (Boundary Layer theory, 1955). It should be evident to the reviewer if her/she is familiar with boundary layer modelling why only an x-direction momentum equation is necessary as it captures the dominant stretching sheet motion (in the axial direction).

  1. 8: We must have a justification for similarity transformations (7), since self-similar solutions greatly simplify the flow structure. Such solutions may not describe the actual physical system.

Reply: Similarity is used to simplify the model from primitive equations via realistic physical scaling. The article is not focused on justification for similarity- neither is the journal! This is not a mathematics journal, it is an engineering applications journal (micromachines). Our work is focused on applying the similarity idea to scale the conservation equations to reduce them to ordinary differential equations. This is a standard practice in boundary layer coating analysis and provides a framework for analyzing physical effects. To describe the actual physical system properly one would require a 3-D computational fluid dynamics model and commercial software e.g. ANSYS FLUENT. This is not our approach. Even commercial software have limitations since they do not feature nanaoscale or advanced viscoelastic models. This is why we have had to use the present approach, which is valid and carefully formulated and solved, as testified to by benchmarking with previous simpler models.  

  1. 9: There is no remarkable finding in the investigation; all those are usual and already obtained previously. In addition, in the “Results and discussion” section the many results are just described (which can be observed by anyone from the figures) without proper physical reasoning

Reply: With due respect this comment is not warranted. The novelty of the present work clearly confirms that new results have been produced which have never been produced before. Important implications of the modelling results have also been included. These are lacking in many comparable articles and there is improved physical reasoning now in the revised discussion.  Please see the new text:  

“Figs. 3-9 disclose the impact of ,   ,    and  on velocity . Fig. 3. demonstrates that an increment in the velocity ratio factor ,  indicates an improvement with respect to η in the boundary regime, that is, the fluid motion is accelerated on the stretchy surface. When  free stream velocity is stronger than the stretching velocity of the surface, this builds a momentum upsurge in the flow domain through exterior free stream which exhibits high acceleration for all horizontal coordinate values, η. Consequently, the thickness of the momentum boundary layer of the stretching surface is therefore decreased. This has an influence on the quality control of manufactured coatings. However, when  the stretching velocity of the sheet is higher than the exterior velocity of free stream and the reverse impact is calculated.  i.e., velocity of Oldroyd-B fluid  is decreased and thickness of momentum boundary layer on the surface increases. If  both the external and the stretchy velocity of the Oldroyd-B fluid are identical. This scenario is a natural intermediate between cases where A > 1 and A< 1. It is evident that higher stretching sheet prevents the development of momentum; however, a stronger external velocity of corresponding Oldroyd-B fluid is produced. The influence of Deborah numbers on the velocity of Oldroyd-B fluid  that is in terms of (  relaxation time quantity) and (  retardation time quantity) are portrayed in Fig. 4 and 5. respectively. Here in Fig. 4, it is clear that as (relaxation time fluid)  rises, the Oldroyd-B fluid velocity gradually declines. In fact, is mathematically expressed as ‘‘the ratio of the observational timescale to the time scale of the material reaction”. We can examine the three different scenarios to assess the polymeric behavior of materials. (i) When  for viscoelastic substance, (ii) when  for entirely viscous material, (iii) when  for elastic material in nature. Higher values of  lead to a lower relaxation relative to the characteristic timescale. This means that fluid react in a similar way to solid materials. Fig. 5 displays the impact of retardation time  parameter on Oldroyd-B fluid velocity . As anticipated,  upsurges when  is increased. Physically, retardation time augments for increasing  owing to which  upsurges. Fig. 6 reveals the features of mixed convection variable  on . Clearly  increases when the mixed convection variable is elevated. Since thermal buoyancy forces exceed the viscous forces with higher values of , this  intensifies the flow. Fig. 7 unveils the impact of buoyancy ratio variable on . It can be seen that a massive boost is induced in the linear velocity across the domain with stronger values of . Note that  signifies that the concentration buoyancy force   of nanoparticles is much stronger than the temperature buoyancy force . Consequently, the hydrodynamic boundary layer thickness of Oldroyd-B fluid increases. Fig. 8 depicts the characteristics of wall suction/injection on . It is analyzed that the impetus of linear velocity is increased with amplification in the injection parameter . On the other hand, it is decreased for greater values of wall suction factor . It is evident that injection of nanofluid augments momentum development in the boundary layer regime and, consequently, velocity of Oldroyd-B nanofluid increases. This effect is also known as blowing in manufacturing processes. The reverse situation is noticed for suction which causes the boundary layer to adhere more strongly to the wall and decelerates flow- this increases momentum boundary layer thickness.  

6.2. Temperature Profile

Figs.9–14 display the effect of          and  on . Fig. 9 depicts the  curves subjected to Prandtl number  values. This Fig. exhibits that the nanofluid temperature decreases for larger Prandtl number. Prandtl number ( ) communicates the rate of momentum to thermal diffusion. Liquid conductivity retards for larger  The heat transferred via conduction of molecules is subsequently repressed which expresses a decay in  and a diminution in thickness of thermal boundary-layer. Chilling of stretched coating system is thus attained with increasing  while heating is witnessed with lower . Fig. 10 illustrates the curves of  under the impact of . It is witnessed that when is increased, is obviously enhanced. Actually, greater heat (thermal energy) is produced in the working liquid during the radiation process (i.e., for higher R values), causing the temperature of the nanofluid in the boundary layer regime to increase. Fig. 11 divulges the consequence of Buongiorno's model parameter (nanoscale thermophoresis ) on thermal distribution  of Oldroyd-B fluid. Stronger movement of nanoparticles in the stagnation boundary layer flow domain is encouraged by the thermophoretic body force; nanoparticles are mobilized  from the hotter region to the cooler one and therefore higher thermal transmission arises in the flow field. Hence, intensifying magnitude of nanoscale thermophoresis parameter  produces substantial thermal diffusion enhancement and boosts temperature and thermal boundary layer thickness. The influence of the Brownian motion factor  on the temperature  of the Oldroyd-B nanofluid are presented in Fig. 12. There is an accentuation in nanofluid temperature  and also thermal boundary-layer thickness with increment in . In fact, the random movement of the fluid particles is increased since nanoparticle diameters are reduced with higher Brownian motion parameter and this generates intensification in ballistic collisions which produces extra heat in the regime. Micro-convection around the nanoparticles is also enhanced with greater Brownian motion effects. All these factors contribute to marked thermal enhancement.The effect of temperature ratio variable  on temperature distribution  is illustrated in Fig. 13. Physically, temperature appears to rise significantly as  increases. Higher values of  implies an elevation in wall temperature which makes the depth of thermal penetration deeper into the boundary layer. Heat transfer into the flow from the wall is therefore encouraged. Also, when the liquid temperature  exceeds the ambient temperature  (in energy equation) this creates a larger temperature differential across the boundary layer which intensifies thermal diffusion from the wall to the free stream and manifests in a boost in  rises. Fig. 14 shows that with larger values of thermal Biot number , there is an improvement in the temperature field . The condition  suggests the configuration of isoflux at the wall, while  symbolizes the configuration of an isothermal wall. In addition, this parameter is featured in the prescribed boundary condition   (from Eqn. (11)) which determines the intensity of Biot number. Therefore, stronger values of Biot number  corresponds to an amplification in thermal convection over the stretching sheet which enhances the nanofluid temperature. The inclusion of this complex convective wall boundary condition provides a more realistic estimation of manufacturing conditions than conventional boundary conditions.   

6.3. Concentration Profile

Figs.15–19 show the impact of   ,    and on . Fig. 15 reveals that with higher values of  the nanoparticle concentration  and thickness of solutal boundary-layer reduces. Actually, Schmidt number is the ratio of momentum to the nanoparticle molecular diffusivity which implies that when rises, mass diffusivity reduces and there is a depletion in concentration . The judicious selection of nanoparticles to embed in the coating regime is therefore critical in achieving bespoke mass transfer characteristics in fabrication processes. Figs. 16 and 17 portray the evolution in concentration   of nanoparticles with different values of thermophoresis  and Brownian   diffusion parameters. When  increases, an increasing trend is seen via higher chaotic movement associated with higher , the particle collision is boosted and mass diffusion is assisted. As a result, the Oldroyd-B nanofluid concentration upsurges. The reverse trend is noted for the influence of thermophoresis parameter  on the surface concentration . Physically, the movement of nanoparticles from the wall to the interior boundary layer region is impeded by an elevation in the thermophoretic force. Therefore, this factor leads to reduction of  i. e. mass diffusion into the boundary layer is suppressed. Fig. 18 displays that with increasing values of mass transfer Biot number , there is an enhancement in magnitudes of the concentration field . This key parameter appears in the relevant boundary conditions of concentration.  i.e.  . Greater values of  significantly boost the concentration of nanoparticles but decrease the gradient of concentration at the wall sequentially. In Fig. 19, it is evident that higher values of destructive parameter , the concentration profile of Oldroyd-B nanofluid exhibits a diminishing trend. With larger values of generative parameter , the concentration of nanoparticles increases.  Practically, it is found that when the reaction rate in the fluid is increased, then greater conversion of the original nanoparticle species is induced in the presence of chemical reaction and thus the concentration of the original nanoparticles reduces. On the other hand, when the reaction rate in the liquid is decreased, then less original species is converted and nanoparticle concentration values are increased.

The numerical findings for and  for diverse values of , , , ,   and  are disclosed in Table. 4. Here it is examined that, the heat and mass transfer rates suppressed for the growing values of relaxation  and retardation  fluid parameters, respectively. Since the stronger values of  and  upsurge the relaxation and retardation times parameters of Oldroyd-B liquid, this induces a reduction in rate of heat and mass transfer at the wall. The heat and mass transfer rates are boosted for greater values of . When the liquid temperature  is greater than the ambient temperature  this effectively increases the thermal conductivity of nanofluid, as  increases. Heat transfer to the wall is therefore enhanced. Moreover, the chemical reaction variable  increasing suppresses concentration and temperature magnitudes in the boundary layer but elevates transport of nanoparticles and heat to the wall.  and  are therefore increased. It is also apparent that heat and mass transfer rates are suppressed with increasing values of Brownian and thermophoretic parameters. This is consistent with the results described earlier wherein temperatures and nanoparticle concentrations were boosted with these parameters. This leads to a depletion in the migration of heat and nanoparticle species to the wall away from the boundary layer and explains the plummet in local Nusselt and Sherwood number functions with higher Brownian motion and thermophoresis parameters. Again the prescription of appropriate nanoparticles is critical in developing the desired nano-coating properties for delicate micro-machining applications since heat, mass and momentum characteristics are strongly influenced by for example nanoparticle mass diffusivity, nanoparticle thermal conductivity and also the nanofluid viscosity which is a function of the nanoparticle concentration.”

  1. 10: Please discuss at the end of introduction, that why is it important from the aspect of APPLICATION and future experiments to investigate the current study.

Reply: We have explained in the revised version the application. See the end of the introduction:

“The applications of the present non-Newtonian nanofluid stagnation model includes coating manufacturing processes for biomimetic sensors [51], optical fiber nanocoatings [52,53] and micro-robot surface protection [54, 55]. The key objective of the present study is to simultaneously consider multiple effects which feature in real manufacturing stagnation flows for nanopolymer coatings including complex thermal convective wall boundary conditions, rheology, high temperature (radiative flux), Brownian motion and thermophoresis. These have not been addressed previously with the Oldroyd-B viscoelastic model. “

  1. 11: The units of the parameters can be included.

Reply: It is done.

  1. 12: In the last paragraph of the introduction section authors are requested to mention all the additional parameters considered in this work. The ones which are not already reported in the literature. Justification must be given why these additional parameters are important.

Reply: we have done this- see:

The applications of the present non-Newtonian nanofluid stagnation model includes coating manufacturing processes for biomimetic sensors [51], optical fiber nanocoatings [52,53] and micro-robot surface protection [54, 55]. The key objective of the present study is to simultaneously consider multiple effects which feature in real manufacturing stagnation flows for nanopolymer coatings including complex thermal convective wall boundary conditions, rheology, high temperature (radiative flux), Brownian motion and thermophoresis. These have not been addressed previously with the Oldroyd-B viscoelastic model.

  1. 13: The paper didn't include the application background of the present paper at the end of introduction.

Reply: As answered earlier, see:

The applications of the present non-Newtonian nanofluid stagnation model includes coating manufacturing processes for biomimetic sensors [51], optical fiber nanocoatings [52,53] and micro-robot surface protection [54, 55]. The key objective of the present study is to simultaneously consider multiple effects which feature in real manufacturing stagnation flows for nanopolymer coatings including complex thermal convective wall boundary conditions, rheology, high temperature (radiative flux), Brownian motion and thermophoresis. These have not been addressed previously with the Oldroyd-B viscoelastic model.

  1. 14: In the introduction section, the authors should cite the drawbacks of earlier studies in each paragraph. This would highlight how the current work is different from the existing works and the advantages of the chosen analysis as well.

Reply: Again we have explained why the current work is different from previous studies- see:

The applications of the present non-Newtonian nanofluid stagnation model includes coating manufacturing processes for biomimetic sensors [51], optical fiber nanocoatings [52,53] and micro-robot surface protection [54, 55]. The key objective of the present study is to simultaneously consider multiple effects which feature in real manufacturing stagnation flows for nanopolymer coatings including complex thermal convective wall boundary conditions, rheology, high temperature (radiative flux), Brownian motion and thermophoresis. These have not been addressed previously with the Oldroyd-B viscoelastic model.

  1. 15: The conclusions must be More Specific of the obtained results. The present conclusions are more general. This section must be revised and authors need to state their key findings. If possible, they can quantify the results in terms percentage of variations with respect to each parameter and this would provide a clear picture of each of the parameters studied. Also, setbacks or limitations of present work can be stated at the end.

Reply: The conclusions have been revised and are focused on implications also of the current modelling. There is no need for quantifiying results in terms of percentages. The numerical section is adequate. The whole purpose of conclusions is to show the key findings, not to include any further numerical data! Please see-

Inspired by analyzing more rigorously the stagnation-flow coating regimes in micro-machine and robotic manufacturing, the current article has developed a novel mathematical model for nonlinear mixed convection flow of Oldroyd-B type nanoliquid subjected to a chemical reaction from the stretching surface. The impacts of thermophoresis along with Brownian motion features induced by nanoparticles are investigated using Buongiorno's nanoscale model. High-temperature heat transfer is evaluated considering a nonlinear version of thermal radiation. The consistent momentum, energy, and nanoparticles volume fraction equations are altered into a set of nonlinear ordinary differential ones and then solved analytically utilizing the homotopy algorithm. Comparison of HAM results with earlier studies are included. The assessment of convergent series solutions is also presented. The main findings emerging from the elaborated analysis are summarized below:

  1. Influences of fluid relaxation and retardation times parameters are qualitatively opposite for the velocity profile of the Oldroyd-B nanoliquid (coating).
  2. Increment in mixed convection parameter improves the velocity profile.
  3. Temperature and nanoparticles concentration show an increasing trend for higher values of thermophoresis parameter.
  4. A rise in temperature ratio parameter leads to higher temperature magnitudes.
  5. Stronger values of thermal and solutal Biot numbers cause an augmentation in fluid temperature and concentration of nanoparticles.
  6. An enhancement in Brownian motion transfers kinetic energy to heat energy, which increases the temperature of the nanofluid and elevates thermal boundary layer thickness.
  7. The current Oldroyd-B viscoelastic nanofluid model reduces to the classical viscous fluid case when . Important thermal, hydrodynamic and species transport characteristics can be captured with the Oldroyd-Bviscoelastic formulation presented which are not possible with classical Newtonian or simpler non-Newtonian models. The results presented are more realistic for actual nano-polymeric coating processes in robotics, sensor and micromachine surface deposition technologies.

The current study provides a good platform for further research in high temperature nano-polymeric coating flow processes. It has however neglected certain aspects including consideration of variable sheet thickness, activation energy, Joule heating, entropy generation, Cattaneo-Christov heat-mass fluxes and bio-convective flows with gyrotactic microorganisms. These may be explored imminently.

  1. 16: Authors need to restructure the results section by giving more importance to physical interpretation rather than describing the graphs. This would explore the important outcomes obtained in this analysis.

Reply: We have done this- see the revised discussion: (we have answered your query earlier above)

Round 2

Reviewer 2 Report

Paper can be accepted in present form